# Microglia and Astroglia—The Potential Role in Neuroinflammation Induced by Pre- and Neonatal Exposure to Lead (Pb)

**DOI:** 10.3390/ijms24129903

**Published:** 2023-06-08

**Authors:** Magdalena Gąssowska-Dobrowolska, Mikołaj Chlubek, Agnieszka Kolasa, Patrycja Tomasiak, Jan Korbecki, Katarzyna Skowrońska, Maciej Tarnowski, Marta Masztalewicz, Irena Baranowska-Bosiacka

**Affiliations:** 1Department of Cellular Signalling, Mossakowski Medical Research Institute, Polish Academy of Sciences, Pawińskiego 5, 02-106 Warsaw, Poland; mgassowska@imdik.pan.pl; 2Department of Biochemistry and Medical Chemistry, Pomeranian Medical University in Szczecin, Powstańców Wlkp. 72, 70-111 Szczecin, Poland; mikolaj.chlubek@gmail.com (M.C.); jan.korbecki@onet.eu (J.K.); skowronska.kt@gmail.com (K.S.); 3Department of Histology and Embryology, Pomeranian Medical University in Szczecin, Powstańców Wlkp. 72, 70-111 Szczecin, Poland; agnieszka.kolasa@pum.edu.pl; 4Department of Physiology in Health Sciences, Pomeranian Medical University in Szczecin, Żołnierska 54, 70-210 Szczecin, Poland; patrycja.tomasiak@pum.edu.pl (P.T.); maciejt@pum.edu.pl (M.T.); 5Department of Anatomy and Histology, Collegium Medicum, University of Zielona Góra, Zyty 28 St., 65-046 Zielona Góra, Poland; 6Department of Neurology, Pomeranian Medical University in Szczecin, Unii Lubelskiej 1, 71-252 Szczecin, Poland; marta.masztalewicz@pum.edu.pl

**Keywords:** lead (Pb), neurotoxicity, microglia, ionized calcium-binding adapter molecule 1 (Iba1), M1 and M2 phenotype of microglia, astrocyte, glial fibrillary acidic protein (GFAP), chemokines (CXCL1, CXCL2), glutamine synthase (GS), brain

## Abstract

Neuroinflammation is one of the postulated mechanisms for Pb neurotoxicity. However, the exact molecular mechanisms responsible for its pro-inflammatory effect are not fully elucidated. In this study, we examined the role of glial cells in neuroinflammation induced by Pb exposure. We investigated how microglia, a type of glial cell, responded to the changes caused by perinatal exposure to Pb by measuring the expression of Iba1 at the mRNA and protein levels. To assess the state of microglia, we analyzed the mRNA levels of specific markers associated with the cytotoxic M1 phenotype (*Il1b, Il6,* and *Tnfa*) and the cytoprotective M2 phenotype (*Arg1, Chi3l1, Mrc1, Fcgr1a, Sphk1,* and *Tgfb1*). Additionally, we measured the concentration of pro-inflammatory cytokines (IL-1β, IL-6, and TNF-α). To assess the reactivity and functionality status of astrocytes, we analyzed the GFAP (mRNA expression and protein concentration) as well as glutamine synthase (GS) protein level and activity. Using an electron microscope, we assessed ultrastructural abnormalities in the examined brain structures (forebrain cortex, cerebellum, and hippocampus). In addition, we measured the mRNA levels of *Cxcl1* and *Cxcl2*, and their receptor, *Cxcr2*. Our data showed that perinatal exposure to Pb at low doses affected both microglia and astrocyte cells’ status (their mobilization, activation, function, and changes in gene expression profile) in a brain-structure-specific manner. The results suggest that both microglia and astrocytes represent a potential target for Pb neurotoxicity, thus being key mediators of neuroinflammation and further neuropathology evoked by Pb poisoning during perinatal brain development.

## 1. Introduction

Accumulating evidence from a combination of epidemiological studies and animal research strongly suggests that environmental factors play a significant role in increasing the susceptibility to neurological and neurodegenerative diseases, particularly when individuals are exposed during the critical pre- and neonatal developmental stages [1]. Lead (Pb), a heavy metal with well-known neurotoxic effects, holds the undesirable distinction of being ranked as the second most toxic substance on the Agency for Toxic Substances and Disease Registry’s priority list in 2017 [2]. Its detrimental impact on the nervous system is especially pronounced when exposure occurs during the crucial periods of prenatal and neonatal development [3,4,5,6,7]. Previously, a blood Pb concentration of 10 μg/dL was considered a safe threshold [8,9]. However, recent research has convincingly demonstrated that even lower levels of blood Pb can lead to neurobehavioral deficits [10,11,12]. As a result, the currently recommended “threshold Pb level” for children and pregnant women has been revised to 5 μg/dL [13]. One of the proposed mechanisms underlying the neurotoxicity of lead (Pb) is its capacity to induce inflammation [14,15,16].

The stimulation of neuroinflammation in response to pathological insults serves as a protective mechanism that is initially designed to protect the brain by removing or inhibiting various pathogens and promoting tissue repair to achieve a state of equilibrium [17]. Nevertheless, enduring or persistent inflammatory reactions encompass chronic activation of glial cells and the subsequent release of diverse pro-inflammatory substances, ultimately impeding regeneration processes and precipitating neuronal cell demise [18,19]. Prolonged neuroinflammation can ultimately result in neurotoxicity and is related to both neurodegeneration and neurodevelopmental diseases [20,21,22].

Microglia and astrocytes play a crucial role in the development of inflammatory processes in the brain [17]. These two primary types of glial cells play a pivotal role in regulating the immune response to pathological processes in the brain [23]. The bidirectional interaction between microglia and astrocytes modulates central nervous system (CNS) inflammation [24]. Acting as the principal innate immune cells in the brain, microglia assume the crucial responsibility of being the frontline defense against pathological alterations within the central nervous system (CNS) [25,26,27]. Investigations on peripheral macrophages have unveiled the existence of distinctive activation profiles for microglia, commonly known as “classical” (M1) and “alternative” (M2) activation states [25,28]. The expression of M1 markers, such as interleukin 1β (IL-1β), interleukin 6 (IL-6), tumor necrosis factor α (TNF-α), interferon γ (IFN-γ), cluster of differentiation 86 (CD86), inducible nitric oxide synthase (iNOS), and oxidative and nitrosative free radicals (such as superoxide, nitric oxide, peroxynitrite), alongside chemokines such as CXCLs and CCLs, indicates the activation of “classical” microglia associated with pro-inflammatory and cytotoxic functions. These activated microglia can instigate a wide range of inflammatory reactions and potentially contribute to impaired neurogenesis and dysfunction of the neurotrophic system [29]. On the other hand, the expression of M2 markers, such as arginase-1 (Arg1), chitinase-like-3 protein 1 (CHI3L1), mannose receptor C-type 1 (MRC1), interleukin 10 (IL-10), Fc-gamma receptor Ia (FCGR1A), transforming growth factor β1 (TGF-β1), and sphingosine kinase 1 (SphK1), indicates “alternative” activation of the anti-inflammatory, neuroprotective phenotype of microglia that counteract the inflammatory-induced damages in CNS [30,31,32,33,34,35].

Astroglia play a crucial role in maintaining the homeostasis of the nervous system and the brain as an organ [17,26,36,37,38]. One of the key functions of astrocytes is controlling neurotransmitter homeostasis, with glutamate homeostasis being of particular importance, as this is critical to both normal brain function, such as neurodevelopment and synaptic plasticity, and pathological processes, such as excitotoxic cell death [39]. Astrocytes occupy a strategic position around synapses and express glutamate transporters and glutamine synthetase (GS) activity [39]. Glutamine serves as a precursor for important neurotransmitters such as glutamate and gamma-aminobutyric acid (GABA), and its availability is dependent on the metabolic activities and functional state of astrocytes [39,40,41,42,43,44,45,46,47].

Similar to microglial activation, astrocyte activation is a heterogeneous process and can be broadly categorized as either neurotoxic (A1-phenotype astrocytes) or neuroprotective and immunoregulatory (A2-phenotype astrocytes) [48,49]. Pro-inflammatory astrocytes release harmful pro-inflammatory factors such as IL-1β, TNF-α, IL-6, and NO, which can have detrimental effects. Conversely, neuroprotective reactive astroglial cells secrete a multitude of beneficial neurotrophic factors and thrombospondins [48]. Astrocytes possess the capability to switch their phenotype and undergo proliferation in response to certain damaging conditions. The proliferation of these transformed (“reactive”) astrocytes within brain tissue is referred to as reactive astrogliosis. This reactive astrogliosis is observed in various pathological conditions, including intracranial infections, hypoxia/ischemia, Alzheimer’s disease, and epilepsy. Alterations in the molecular expression and morphology of astrocytes, as indicated by the glial fibrillary acidic protein (GFAP), can serve as indicators of the severity of reactive astrogliosis, which is a significant hallmark of multiple CNS pathologies [50]. GFAP, a major component of astrocyte intermediate filaments, is commonly employed as a marker for astrocytes [49]. Upregulation of GFAP serves as an early indication of CNS injury [24,51]. To initiate a secondary inflammatory response, pro-inflammatory astrocytes are activated by inflammatory mediators released by activated M1 microglia, including IL-1β, IL-1α, TNF-α, and complement component 1q (C1q) [48,52]. IL-1 plays a crucial role as a mediator, amplifying the secretion of other cytokines, particularly IL-6, mainly from astrocytes, thereby promoting inflammation [53,54,55]. Elevated levels of pro-inflammatory cytokines, such as IL-1β and TNF-α, stimulate an upregulation of chemokines, including CXCL1 and CXCL2, along with their primary receptor CXCR2 [56,57,58,59,60,61,62,63].

The effect of Pb on the processes of induction and propagation of inflammation in the CNS is well documented. However, the precise molecular mechanisms responsible for the pro-inflammatory properties of Pb in the brain have not been fully investigated. Furthermore, no data are available on the effect of pre- and neonatal exposure to low-dose Pb on the glial cell status/profile.

Therefore, the aim of our study was to investigate the involvement of glial cells, especially microglial and astroglial cells, in the neuroinflammation induced by perinatal exposure to Pb. In our experimental conditions, an analysis of the genetic markers of M1 and M2 was performed to explore the phenotype polarization of microglial activation. We analyzed the expression of the selected pro-inflammatory cytokines (*Il1b*, *Il6*, and *Tnfa*), which are markers of the cytotoxic phenotype M1 microglial activation, as well as the selected markers of the M2 phenotype (*Arg1, Chi3l1, Mrc1, Il10, Fcgr1a, Sphk1*, and *Tgfb1*). In the case of M1 markers, in addition to the expression at the mRNA level, the concentration of pro-inflammatory cytokines (IL-1β, IL-6, and TNF-α) was also determined. Moreover, a marker that stains both resting and activated microglia, Iba1 (mRNA expression and protein concentration) was evaluated. Additionally, to appraise the reactivity and functionality status of astroglia, the *Gfap* mRNA expression, the concentration of GFAP protein, and both GS protein level and activity were analyzed. The ultrastructure of glial cells in the hippocampus, forebrain cortex, and cerebellum was also examined using the TEM method. Moreover, the mRNA levels of selected chemokines, *Cxcl1* and *Cxcl2*, and their receptor, *Cxcr2*, in the brain of rats’ offspring exposed to Pb were measured. Our data revealed that pre- and neonatal exposure to Pb at low doses affects both microglia and astrocyte cells’ status (their mobilization, activation, and changes in the gene expression profile) in a brain-structure-specific manner. In the forebrain cortex and cerebellum, both the pro-inflammatory M1 and the potentially beneficial recovery-promoting M2 phenotype were activated, while in the hippocampus, only the classic M1 was stimulated. Additionally, we revealed that astrocyte activation, as well as disturbances in the astrocytes’ GS protein level and activity were results of neurotoxic Pb action. Moreover, our results suggest that the CXCL1/CXCR2 and CXCL2/CXCLR2 pathways may be involved in Pb-induced brain pathology in rat offspring during pro-inflammatory microglia and astrocyte activation. All these results suggest that both microglia and astroglia may represent a potential target for Pb toxicity, thus being key mediators of neuroinflammation and further neuropathology induced by Pb administration during perinatal brain development.

## 2. Results

### 2.1. Perinatal Exposure to Pb Increased Lead Concentration in Whole Blood and Brain Tissue

The lead exposure protocol utilized in our study, starting from the first day of fetal life and continuing through maternal feeding until postnatal day 21 (PND), resulted in a statistically significant elevation in the concentration of lead (Pb) in whole blood, measured at PND 28. The final levels reached 6.59 ± 0.11 μg/dL in the study group compared to 0.05 ± 0.10 μg/dL in the control group, demonstrating a significant difference (*p* = 0.002).

Furthermore, the Pb concentration in all examined brain structures was significantly higher in the study group compared to the control group. The highest concentration was observed in the hippocampus for both the study group (7.49 ± 0.26 µg/dL) and the control group (0.29 ± 0.32 µg/dL) (*p* = 0.001). This was followed by the cerebellum (7.43 ± 0.11 µg/dL vs. 0.02 ± 0.01 µg/dL) and the forebrain cortex (7.22 ± 0.16 µg/dL vs. 0.02 ± 0.05 µg/dL), with statistically significant differences observed between the study and control groups (*p* = 0.001) for both brain structures.

The levels of Pb in the brain showed a strong positive correlation with the levels of Pb in whole blood, as evidenced by the correlation coefficients (Rs) for the forebrain cortex (+0.65), cerebellum (+0.65), and hippocampus (+0.85) (*p* < 0.005 for all brain structures examined).

### 2.2. Perinatal Exposure to Pb Affected the Microglia in the Brain of Adult Rats

To assess the response of microglia to the changes induced by perinatal exposure to lead (Pb), we examined the gene expression and concentration of ionized calcium-binding adapter molecule 1 (Iba1). Iba1 is a calcium-binding protein expressed by both surveillant and activated microglia. Furthermore, we analyzed the mRNA levels of specific pro-inflammatory cytokines (markers of the cytotoxic microglia activation phenotype M1, including *Il1b*, *Il6*, and *Tnfa*), as well as markers associated with the cytoprotective M2 phenotype (such as *Arg1*, *Chi3l1*, *Mrc1*, *Il10*, *Fcgr1a*, *Sphk1*, and *Tgfb1*). These additional evaluations provided insights into the status and functional characteristics of microglial cells.

The quantitative real time polymerase chain reaction (qRT-PCR) analysis demonstrated a significant increase in the mRNA levels of *Iba1* in all examined brain structures following perinatal exposure to lead (Pb), as illustrated in Figure 1. Notably, the most substantial elevation in *Iba1* gene expression was observed in the forebrain cortex, where it increased by approximately 160% (*p <* 0.0001) compared to the corresponding control group (Figure 1(B.1)). Similarly, in the hippocampus, the mRNA level of *Iba1* showed a significant increase of nearly 140% (*p <* 0.0001) (Figure 1(A.1)). In the cerebellum of Pb-treated rats, the mRNA coding for *Iba1* exhibited an approximately 130% increase (*p <* 0.0001) compared to the control group (Figure 1(C.1)). These changes in *Iba1* mRNA levels were further validated by an immunoenzymatic (ELISA) test. Our results revealed a significant (*p <* 0.0001) increase in the concentration of Iba1 by approximately 100%, 68%, and 99% in the hippocampus, forebrain cortex, and cerebellum, respectively (Figure 1(A.2,B.2,C.2)). The ELISA results were confirmed by immunohistochemical analysis, which revealed significantly higher immunoexpression of Iba1 in all examined brain structures of Pb-treated rats (Figure 1D,D’–F’) than in control animals (Figure 1D,A’–C’). In the control (Figure 1D,A’) and Pb-treated (Figure 1D,D’) rats the Iba1-positive cells (microglia cells, brown-colored cells) were present in the neuropil of the hippocampus between nerve cells of the Gyrus Dentate (GD) and Cornu Ammonis (both structures visible), and Iba1-immunopositive cells were much more numerous after Pb-intoxication (Figure 1D,D’). In the neuropil of the forebrain cortex of control rats (Figure 1D,B’), the Iba1-reactive cells were occasionally visible in contrast to Pb-treated rats (Figure 1D,E’), in which the Iba1-immunopositive cells were more prominent in all six layers of the gray matter. In the cerebellum, Iba1-positive cells were mainly in the Granular Cell Layer of gray matter and in the white matter, however, in Pb-treated rats, the immunoexpression of Iba1 in these areas of the cerebellum was significantly higher (much more microglia cells were visible, compared to control) (Figure 1D,C’,F’).

Additionally, prenatal and neonatal exposure to Pb resulted in a significant increase in the gene expression of the pro-inflammatory cytokines *Il1b*, *Il6*, and *Tnfa* in the hippocampus, forebrain cortex, and cerebellum, as illustrated in Figure 2, Figure 3 and Figure 4(A.1,A.3,A.5). Specifically, compared to the control groups, the expression of *Il1b*, *Il6*, and *Tnfa* was significantly elevated in the hippocampus by approximately 144% (*p* < 0.0001), 163% (*p* < 0.0001), and 79% (*p* < 0.0001), respectively (Figure 2(A.1,A.3,A.5)). In the forebrain cortex, the mRNA levels of *Il1b, Il6*, and *Tnfa* increased by approximately 84% (*p* = 0.0005), 100% (*p* = 0.0029), and 40% (*p* < 0.0001), respectively, in response to Pb exposure (Figure 3(A.1,A.3,A.5)). Similarly, in the cerebellum of Pb-treated rats, a significant increase in the gene expression of *Il1b* (by approximately 210%, *p* < 0.0001), *Il6* (by approximately 206%, *p* < 0.0001), and *Tnfa* (by approximately 57%, *p* = 0.0384) was observed compared to the control groups (Figure 4(A.1,A.3,A.5)).

In addition to the increased gene expression of the pro-inflammatory cytokines, a significant increase in the concentration of IL-1β, IL-6, and TNF-α was also observed (Figure 2, Figure 3 and Figure 4(A.2,A.4,A.6)). IL-1β exhibited a significant increase of 100% in the forebrain cortex (*p* = 0.0020), 93% in the cerebellum (*p* < 0.0001), and a non-significant increase of 40% in the hippocampus (Figure 2, Figure 3 and Figure 4(A.2)). IL-6 increased by 74% in the hippocampus (*p* = 0.0212), 63% in the forebrain cortex (*p* = 0.0029), and showed a non-significant increase of 22% in the cerebellum (Figure 2, Figure 3 and Figure 4(A.4)). The concentration of TNF-α showed the greatest increase of 110% in the hippocampus (*p* = 0.0007), followed by a 90% increase in the cerebellum (*p* = 0.0020), and a 76% increase in the forebrain cortex (*p* < 0.0001) (Figure 2, Figure 3 and Figure 4(A.6)).

These findings collectively suggest the activation of localized pro-inflammatory responses associated with microglia in the brains of offspring rats due to perinatal exposure to Pb.

In turn, the analysis of the markers of the M2 phenotype revealed a lack of changes in the gene expression of all analyzed markers of alternative, anti-inflammatory, and neuroprotective M2 phenotypes of microglia (M2a, M2b, and M2c subtypes) in the hippocampus of the Pb-treated offspring (Figure 2(B.1–B.3)). Additionally, in the forebrain cortex, we observed a very similar response of the M2 phenotype. With the exception of *Chi3l1* and *Mrc1*, in which the mRNA levels were significantly increased (by about 238% (*p* = 0.0127), and by about 116% (*p* < 0.0001), respectively), the gene expression of *Arg1*, *Il10*, *Fcgr1a*, *Tgfb1*, and *Sphk1* were unchanged, compared to the control. Similarly, in the cerebellum of Pb-exposed rats, the gene expression of most of the analyzed markers of M2a, M2b, and M2c subtypes was unaffected. The exception was mRNA level of *Chi3l1*, which increased by about 151% (*p* = 0.0285), versus the control group.

### 2.3. Perinatal Exposure to Pb Increased the mRNA Expression of Cxcl1 and Cxcl2 Chemokines and the Receptor Cxcr2 in the Brain of Adult Rats

In our study, we observed that Pb exposure had an impact on not only the mRNA levels and concentrations of pro-inflammatory cytokines in the brain of rat offspring, but also on the mRNA expression of certain chemokines, such as chemokine (C-X-C motif) ligand 1 (Cxcl1), chemokine (C-X-C motif) ligand 2 (Cxcl2), and their C-X-C Motif Chemokine Receptor 2 (Cxcr2) in a brain-structure-dependent manner, as depicted in Figure 5. During the inflammatory response, there is an upregulation of pro-inflammatory cytokines, and previous studies have shown that an increased expression in IL-1β and TNF-α can enhance the transcriptional expression of Cxcl1. Our findings indicated that perinatal exposure to Pb resulted in a significant increase in the gene expression of *Cxcl1* in both the hippocampus and forebrain cortex, with approximately 190% (*p* = 0.0002) and 156% (*p* = 0.0335) elevations, respectively (Figure 5A,C). However, no significant changes were observed in the mRNA levels of *Cxcl1* in the cerebellum (Figure 5E). Moreover, Pb also affected the expression of *CxCl2*. Compared to the control group, the gene expression of *Cxcl2* was significantly increased in all analyzed brain structures (Figure 5B,D,F). The levels of mRNA for *Cxcl2* increased most (by 114%, *p* = 0.0002) in the hippocampus, followed by the cerebellum (by 69%, *p* = 0.0002) and the forebrain cortex (by 66%, *p* < 0.0001). In the subsequent analysis, we examined the alterations in the gene expression of C-X-C Motif Chemokine Receptor 2 (Cxcr2). Cxcr2 is a G-protein-coupled receptor that is activated by CXC chemokines, including CXCL1 and CXCL2. The interaction between CXCR2 and its ligands plays a crucial role in facilitating the migration of neutrophils to sites of inflammation.

Our study revealed significant increases in the expression levels of *Cxcr2* in the rat hippocampus (approximately 250%, *p* < 0.0001) and forebrain cortex (approximately 70%, *p* = 0.0127) following perinatal exposure to Pb (Figure 5G,H). Notably, the mRNA levels of *Cxcr2* in the cerebellum remained unchanged (Figure 5I). These findings highlight the brain-region-specific effects of Pb exposure on the gene expression of *Cxcr2*, suggesting a potential involvement of CXCR2-mediated mechanisms in modulating the inflammatory response in the hippocampus and forebrain cortex.

### 2.4. Perinatal Exposure to Pb Affects Astrocytes Phenotype and Function in the Brain of Adult Rats

Considering the crucial role of astrocytes in various developmental processes, any disruptions to their function during early brain development can have detrimental effects on central nervous system (CNS) function. One prominent response of mature astrocytes to CNS damage is reactive astrogliosis, characterized by an increased presence of reactive astrocytes that exhibit distinct features such as larger size, longer and thicker processes, and elevated levels of glial fibrillary acidic protein (GFAP). Increased GFAP expression is widely recognized as a highly sensitive marker of CNS injury. Therefore, in order to investigate the impact of perinatal Pb exposure on astrocyte phenotype, we examined the transcription and concentration of GFAP, a prototypical marker of astroglial cells.

In our study, we observed a significant increase in *Gfap* mRNA levels in all brain structures analyzed in Pb-exposed rats (Figure 6(A.1,B.1,C.1). Specifically, in the hippocampus, *Gfap* mRNA levels were elevated by 110% (*p <* 0.0001), in the forebrain cortex by approximately 87% (*p* < 0.0001), and in the cerebellum by about 97% (*p* < 0.0001), compared to control animals. This increase in *Gfap* gene expression was accompanied by a notable elevation in GFAP protein concentration. The most pronounced increase in GFAP content (approximately 250%, *p* < 0.0001) was observed in the hippocampus (Figure 6(A.2)), followed by the forebrain cortex (approximately 90%, *p* < 0.0001) (Figure 6(B.2)), and the cerebellum (approximately 32%, *p =* 0.0076) (Figure 6(C.2)). The ELISA results were further confirmed by immunohistochemical analysis. Following Pb exposure, the number of astrocytes (GFAP-positive cells) significantly increased, particularly in the hippocampus and cerebellum (Figure 6D,D’–F’), compared to control animals (Figure 6D,A’–C’). In the hippocampus, GFAP-immunopositive cells were more abundant after Pb intoxication, present in the neuropil between nerve cells of the Gyrus Dentate (GD) and Cornu Ammonis (not shown in the microphotography) (Figure 6D,A’,D’). In the forebrain cortex neuropil, GFAP-reactive cells were occasionally visible in both control and Pb-treated rats, but were more prominent in the Pb-treated group (Figure 6D,B’,E’). In the cerebellum, GFAP-positive cells were mainly observed in the Granular Cell Layer of the gray matter and white matter, and the number of astrocytes was significantly higher in the Pb-treated rats’ cerebellum (Figure 6D,C’,F’).

Subsequently, we investigated the impact of pre- and neonatal Pb exposure on the functionality of astrocytes by analyzing the protein level and activity of glutamine synthetase (GS), a crucial enzyme predominantly found in astrocytes. GS is responsible for converting glutamate and ammonia into glutamine in the mammalian brain, playing a vital role in maintaining glutamate and glutamine homeostasis. This process is essential for the synthesis of glutamate and GABA in neurons.

Western blot analysis revealed a significant decrease in GS protein levels in the hippocampus (*p* = 0.0113), with a reduction of approximately 33% (Figure 6E). Similarly, in the forebrain cortex, GS levels were significantly decreased by approximately 28% (*p* = 0.0059) (Figure 6F). Notably, the immunoreactivity of GS in the cerebellum remained unchanged (Figure 6G).

Furthermore, perinatal exposure to Pb resulted in a significant reduction in GS activity across all brain structures examined. In the hippocampus, GS activity decreased by approximately 30% (*p* = 0.0012), while in the forebrain cortex and cerebellum, the reduction was approximately 15% (*p* = 0.0189) and 20% (*p* = 0.0064), respectively (Figure 6H–J). These findings suggest that Pb exposure during fetal development and after birth leads to a decrease in GS protein level and activity, indicating impaired functionality of astrocytes in the regulation of glutamate and glutamine metabolism.

### 2.5. Perinatal Exposure to Pb Causes Ultrastructural Changes in the Glial Cells in the Brain of Adult Rats

Figure 7, Figure 8 and Figure 9 show representative microphotographs of the ultrastructure of control and perinatal Pb exposure rats’ brain tissue. According to our previous study, the ultrastructural pathological alterations in neurons were observed in all analyzed brain structures, however, most strongly were expressed in the hippocampus [5]. In the current study, we examined the ultrastructure of glial cells (microglia and astrocytes) and we observed that the largest pathological changes within these cells caused by Pb intoxication were visible in the cerebellum (Figure 7b–d), such as nuclear protrusions within the nuclear envelope of active astrocytes with an increased number of lysosomes and GFAP, and activation of microglia cells manifesting presence cytoplasmic lysosomes and pinocytotic vacuoles.

The forebrain cortex and hippocampus seem to be less sensitive to the influence of Pb in the case of ultrastructural alterations in glial cells; in the forebrain cortex only activated astrocytes have been observed (Figure 8b,c).

In turn, in the CA1 region of the hippocampus from perinatal Pb exposure rats are visible both active astrocyte with GFAP in the cytoplasm (Figure 9b), and active microglial cells with pinocytotic vacuoles (Figure 9c).

## 3. Discussion

Glial cell activation serves as a prominent marker of neuroinflammatory events, and the release of cytokines and chemokines by these cells plays a role in initiating mechanisms that can result in neuronal loss. The functional activation of microglia and astrocytes, along with subsequent neuroinflammation are closely linked to conditions such as infections, autoimmunity, and the development of neurodegenerative and neurodevelopmental disorders [64].

Our study was carried out using a prenatal and neonatal Pb administration model, which imitates the environmental exposure to Pb in low doses during fetal life and breastfeeding, resulting in blood Pb levels below the previously established “safe” threshold. Nonetheless, our findings demonstrated that both microglia and astrocytes may serve as potential targets for Pb toxicity, thus acting as key mediators of neuroinflammation and further neuropathology induced by Pb during perinatal brain development. Our current report revealed that perinatal Pb exposure induces microglial activation and promotes a pro-inflammatory M1 phenotype in the brains of rat offspring. Specifically, we observed an increased number of mobilized/activated microglia cells, as indicated by raised *Iba1* mRNA expression and concentration, along with significantly higher levels of pro-inflammatory cytokines (IL-1β, IL-6, TNF-α) and chemokines (*Cxcl1*, *Cxcl2* mRNA expression) associated with an increased expression of *Cxcr2* in a brain-structure-dependent manner. The broad spectrum of pathological changes induced by perinatal Pb exposure in the rat brain, as indicated in our previous studies (e.g., oxidative stress with reactive oxygen species (ROS) generation, impairment of pro- and antioxidative balance of neurons, ultrastructural and molecular alterations in synapses, Tau protein pathology, disturbed brain energy metabolism, and neuronal loss) [6,7,65], may be responsible for the strong and the almost exclusively pro-inflammatory response of microglia, particularly in the hippocampus, where only an cytotoxic M1 phenotype was stimulated. The gene expression of all analyzed markers of the M2 phenotype (M2a, M2b, and M2c subtypes), including *Arg1, Chi3l1, Mrc1, Fcgr1a, Tgfb1*, and *Sphk1*, was unaffected in the hippocampus of the offspring perinatally exposed to Pb. The lack of changes in gene expression of the M2 phenotype may reflect the inability of this brain structure to restore homeostasis in response to the wide-scope pathological changes and abnormalities evoked by the exposure to Pb. Changes in the microglia profile are correlated with the type of challenge faced by the CNS. Taking into account the multi-directional neurotoxicity action of Pb in the hippocampus (where oxidative stress and impairment of pro- and antioxidative balance of neurons, as well as a decrease in a number of neurons concomitantly with ultrastructural pathological alterations were the most strongly observed and revealed by us in our previous studies) [4,5,6,7], any attempt to change the microglia phenotype to one that will allow for the repair and reconstruction of damaged tissue in this brain structure turns out to be ineffective and downright impossible. All these results may also suggest a greater sensitivity of the hippocampus to Pb-induced pathology unlike the forebrain cortex or cerebellum. Long-term overactivation of microglia releases various harmful factors which exert stress on neurons, causing a gradual loss of neuronal function or leading to their death. In the hippocampus, we observed the strongest loss of neurons (the number of pyramidal neurons in the CA1 region and granular neurons, as well as the thickness of both the Granular Layer of the Dentate Gyrus and Pyramidal Cell Layer of the hippocampus were significantly lower in rats exposed to Pb compared to the control group) [5].

Our findings suggest that Pb exposure has the potential to worsen CNS damage by promoting M1 polarization of microglia in the hippocampus of rat brains. Additionally, in the forebrain cortex and cerebellum, where pro-inflammatory cytokine expression was significantly altered, perinatal Pb exposure also affected the expression of key markers associated with alternative, anti-inflammatory, and neuroprotective microglia (M2 phenotype). Notably, in a brain-structure-specific manner, there were varying degrees of activation of the potentially beneficial recovery-promoting microglia phenotype M2a, indicating a potential response to Pb-induced abnormalities in the forebrain cortex and cerebellum aimed at brain repair. Up-regulation of the *Chi3l1* and *Mrc1* genes was observed in the forebrain cortex, while only up-regulation of the *Chi3l1* gene was observed in the cerebellum. Thus, in the forebrain cortex, the anti-inflammatory and compensatory response of microglial cells seems to be the strongest. It is plausible that to reestablish homeostasis in the brain, microglial cells in the forebrain cortex and cerebellum, to varying degrees, may have attempted to switch from a pro-inflammatory M1 phenotype to a neuroprotective M2 phenotype during the progression of Pb-induced pathology. As was observed in the forebrain cortex and cerebellum, the attempt to change the microglia phenotype to one that will allow for the repair and reconstruction of damaged tissue can be interpreted in two ways. This response may have been due to the wider range/spectrum of pathological changes or rather the specificity of the Pb-induced changes observed in the forebrain cortex and cerebellum compared to those in the hippocampus (e.g., unlike in the hippocampus, we only observed pathological changes in the Tau protein in the forebrain cortex and cerebellum—its excessive accumulation together with its hyperphosphorylation accompanied with GSK-3β and CDK5 kinases activation). We do not know to what extent Pb-induced hippocampal and cerebellar pathology in the Tau protein may determine phenotypic changes in microglia in these brain structures. In turn, taking into account both the degree of intensity of some pathological changes (oxidative stress and the impairment of pro- and antioxidative balance of neurons, and a decrease in the number of neurons concomitantly with ultrastructural alterations) which were most strongly observed in the hippocampus, and the wider spectrum of some abnormalities associated with synaptic pathology in the hippocampus and cerebellum (dyshomeostasis in a wider range of synaptic proteins, in synaptic vesicle proteins (Syp, Syt1), presynaptic plasma membrane proteins (syntaxin1, SNAP25), and postsynaptic density proteins (PSD95), in contrast to changes in only the PSD95 protein in the forebrain cortex), we can speculate an attempt to stimulate/switch from the M1 to the M2 phenotype in the forebrain cortex and also, to a lesser extent, in the cerebellum; however, it may also be associated with a smaller range or their lesser intensity of pathological changes evoked by Pb in these brain regions. Perhaps, both the nature and the degree of severity of the changes caused by Pb in the forebrain cortex and in the cerebellum made it possible to attempt the activation of the mechanisms of neuroprotection (increase M2 gene expression). Alternatively, all these results may suggest a greater sensitivity of the hippocampus and cerebellum to Pb-induced pathology than the cortex. Nevertheless, all these data indicated that perinatal exposure to Pb induced long-lasting changes in the microglia status in a brain-structure-specific manner. In the forebrain cortex and cerebellum, both a pro-inflammatory and a potentially beneficial recovery-promoting microglia phenotype were activated to a varying extent, while in the hippocampus, only an M1 phenotype was stimulated. To restore homeostasis in the brain, the microglia phenotype in the forebrain cortex and in the cerebellum likely made an attempt to switch from the pro-inflammatory M1 to the neuroprotective M2 during the progression of the pathology. Unfortunately, at this point, we are unable to determine why the microglial response to Pb is specific to the brain regions studied. It is also unknown to what extent the activation of the two or one genes of the M2a phenotype of microglia is able to counteract Pb-induced pathology. However, despite the attempt to activate the M2 phenotype of microglia, we observed a wide range of neurotoxic effects of Pb on the brain (Figure 10), including pathological changes in the forebrain cortex and cerebellum. This evidence suggests rather an inability of microglia to counteract Pb-induced pathology and achieve a state of equilibrium in the brain. Although, in the case of the cortex, the range and intensity of the pathology seem to be limited. Consistent with our findings, previous studies have reported an increased synthesis of pro-inflammatory cytokines in activated microglial cells in various in vivo and in vitro models of Pb exposure [66,67,68,69,70,71]. Liu et al. demonstrated that Pb exposure can induce micro- and astrogliosis in the hippocampus of young mice through the TLR4-MyD88-NF-kB signaling cascade [72]. In a study by Wu et al., in 2021, Pb exposure promoted microglial activation and upregulated the expression of M1 microglial markers while downregulating the expression of M2 markers in the hippocampus of mice with high-fat diets [73]. Mu et al. reported that Pb exposure increased TNF-α levels and significantly affected the expression of 16 genes related to oxidative stress and antioxidant defenses in microglial BV-2 cells [74]. Conversely, Sobin et al., in 2013, did not support a model of increased neuroinflammation in an early chronic Pb exposure model. Instead, they demonstrated that a disruption of microglia occurred in the developing brains of Pb-exposed mice, characterized by damage to, loss of, or lack of proliferation of microglia [75].

Pb poisoning triggers an inflammatory response by involving the collaboration between microglia and astrocytes. Studies have indicated that following CNS insults or during neurodegenerative diseases, astrocytes undergo reactive changes, losing normal functions and adopting abnormal roles that can contribute to pathology [76]. This reactive state, known as “astrocytosis,” is characterized by features such as proliferation, morphological alterations, increased expression of the glial fibrillary acidic protein (GFAP), and changes in gene expression, molecular profile, and metabolism [77]. Elevated levels of GFAP expression serve as a marker for the activation of astrocytes surrounding blood vessels, indicating “mild to moderate” astrogliosis. Studies on systemic inflammation have shown that GFAP transcription increases as early as 6 h after exposure to lipopolysaccharide (LPS) or polyinosinic:polycytidylic acid (poly I:C) [78]. Our study revealed an increase in astrocyte gliosis, as indicated by elevated *Gfap* mRNA expression and protein concentration. We also observed disturbances in the levels and activity of the glutamine synthetase (GS) protein in astrocytes, suggesting an association between prenatal and neonatal Pb exposure and pathological changes in astroglia. Astrocyte activation often leads to a loss of buffering function and contributes to pathological processes. This phenomenon is accompanied by neuronal cell death and may be linked to the production of various cytokines and chemokines. Consistent with our findings, a study involving short-term exposure to high doses of Pb demonstrated glial activation, reflected in increased levels of both GFAP and S-100β proteins in various brain regions [66]. Astrocyte activation in this study was associated with elevated levels of pro-inflammatory cytokines, such as IL-1β and TNF-α in the hippocampus, and IL-6 in the forebrain. Furthermore, Li et al., in 2014, showed that early-life Pb exposure significantly affected GFAP expression in the hippocampus of mouse pups [69]. Additionally, Villa-Cedillo et al., in 2019, demonstrated that chronic Pb treatment induced neurodegeneration, demyelination, and astrogliosis in the rat spinal cord [79].

Glutamine synthetase (GS), an enzyme present in astrocytes, plays a crucial role in converting glutamate and ammonia to glutamine in the mammalian brain. This conversion is essential for maintaining a continuous supply of glutamine, which is necessary for the synthesis of glutamate and GABA in neurons [80]. Our study demonstrated a significant decrease in both the protein level (in the hippocampus and the forebrain cortex) and the activity of GS (in the hippocampus, forebrain cortex, and cerebellum) following perinatal Pb exposure. The observed dysfunction of GS could disrupt the glutamate/glutamine cycling between neurons and astrocytes, leading to disturbances in glutamatergic signaling. Altered expression and activity of GS have been implicated in various neurodegenerative and psychiatric disorders, including Parkinson’s disease (PD), Alzheimer’s disease (AD), epilepsy, schizophrenia, depression, and diabetes [81]. These findings contrast with the results of Strużyńska et al., who reported increased expression of GS and GLAST, along with decreased mRNA expression and GLT-1 protein in the brains of adult rats exposed to high doses of Pb, suggesting that astrocytic processes are involved in the regulation of glutamate homeostasis [66]. In their study, the astroglial response appeared to be neuroprotective, unlike our model of prenatal and neonatal exposure. However, studies by Sierra et al., Robinson et al., and Takeuchi et al. have demonstrated the inhibitory effect of Pb exposure on GS activity in various animal models and cell cultures [41,44,46]. Accumulation of lead leads to reduced GS activity, resulting in glutamate accumulation and excitotoxicity [46]. Furthermore, in accordance with Palmieri et al.’s findings that pharmacological inhibition of GS activity shifts M2-polarized macrophages toward an M1-like phenotype with reduced intracellular glutamine levels [82], the lack of M2 activation observed in the hippocampus in our study could be partly attributed to the decrease in GS level and activity, which was most prominent in this brain region. The GS-catalyzed reaction is also the primary mechanism in the brain for removing ammonia. Excessive ammonia is toxic to cerebral cells through various mechanisms, including oxidative/nitrosative stress resulting from disturbances in the NO pathway, creatine deficiency, and inhibition of the tricarboxylic acid cycle (TCA) [83]. These toxic effects can lead to secondary mitochondrial dysfunction and energy deficits, as demonstrated in our previous study on the same Pb neurotoxicity model [3].

Moreover, in our study, we indicated for the first time that the CXCL1/CXCR2 and CXCL2/CXCLR2 pathways may also be involved in Pb-induced brain pathology in rat offspring during pro-inflammatory glial activation. We revealed a significant increase in the level of mRNA coding for *Cxcl1* and *Cxcl2* in a brain-structure-dependent manner as a result of perinatal Pb exposure. An increased expression of *Cxcr2* has also been shown.

Recent studies have highlighted the association between increased CXCL1 levels in the brains of individuals with Alzheimer’s disease [84]. Upon binding to CXCR2 receptors on neurons, CXCL1 activates signaling pathways such as ERK/MAPK [84] and GSK-3β [85], leading to Tau hyperphosphorylation and subsequent cleavage at Asp421 by caspase-3 upon prolonged exposure to CXCL1 [85]. The truncated Tau protein exhibits enhanced aggregation propensity, potentially initiating the formation of neurofibrillary tangles [85]. Moreover, CXCR2 activation has been shown to upregulate γ-secretase activity [86], which contributes to the cleavage of the amyloid precursor protein (APP) and the release of amyloid-beta (Aβ), thus promoting amyloid plaque formation. These mechanistic insights are particularly relevant in understanding the neurotoxic effects of Pb, as our previous studies have implicated GSK-3β and CDK5-mediated Tau hyperphosphorylation as potential mediators of Pb-induced neuronal dysfunction and cytoskeletal instability [7]. Notably, individuals with Alzheimer’s disease also exhibit increased CXCL1 expression in blood monocytes [87]. In addition, CXCR2 expression has been observed in brain microvascular endothelial cells under the influence of Aβ [87], suggesting a role in promoting transendothelial migration of monocytes into the brain. These monocytes can differentiate into bone marrow-derived microglia, potentially impacting the clearance of Aβ plaques and the progression of Alzheimer’s disease [87]. Given that Pb is known to disrupt the blood–brain barrier, allowing monocytic and macrophagic infiltration into the brain, further investigation is warranted to explore the involvement of these cells in Pb-induced neurotoxicity, as the existing literature on this topic is limited.

In summary, microglia and astrocytes may be the key mediators of neuroinflammation during perinatal brain injury induced by Pb. Our study highlights chronic glial activation and concurrent inflammatory and neurodegenerative features as mechanisms of Pb neurotoxicity in rat brains. Our findings suggest that during neuroinflammation triggered by Pb exposure, microglia are more vulnerable to damage and initially activate into the M1 phenotype, producing typical pro-inflammatory signals such as IL-1β, IL-6, and TNF-α, potentially triggering a reactive astrocyte into the A1 phenotype. Our results suggest that Pb exposure during fetal life and breastfeeding may exacerbate CNS damage by promoting M1 polarization of hippocampal microglia. The microglia switch to the alternatively activated M2 profile only in the forebrain cortex and the cerebellum, although it seems that this is not sufficient for a compensatory response and to counteract Pb-induced damages in the CNS. All these results provide evidence that neuroglial cells may represent a potential target for manipulation in Pb-induced neuroinflammatory brain injury.

## 4. Materials and Methods

### 4.1. Animals—In Vivo Model

The animal procedures conducted in this study adhered to international standards for animal care and welfare, with a focus on minimizing animal use and any potential suffering. Ethical approval was obtained from the Local Ethical Committee for Animal Research at the Pomeranian Medical University in Szczecin (Approval No. 5/2014 of 23 April 2014, annex 2021).

Female Wistar rats, aged three months and weighing 250 ± 20 g (n = 6), were housed for one week with sexually mature males (2:1 ratio) and provided with unrestricted access to food and water. The housing facility maintained a controlled temperature and a 12-h light/dark cycle. After one week, the males were separated, and each female was housed individually. The pregnant females were assigned to either the control or experimental group.

The experimental group (n = 3) received 0.1% lead acetate (PbAc) in their drinking water ad libitum from the first day of pregnancy. The PbAc solution was prepared daily in disposable plastic bags without acidification. The control group (n = 3) received distilled water. There were no significant differences in liquid intake between the two groups. The offspring, comprising both males and females, remained with their mothers and were breastfed. During lactation, the young mothers in the experimental group continued to receive PbAc in their drinking water. The young rats were weaned on postnatal day 21 (PND 21) and housed individually. From PND 21 to PND 28, both the experimental and control groups were provided with only distilled water ad libitum.

The chosen method of lead exposure involved the administration of 0.1% PbAc in drinking water, as it simulates environmental exposure and is a commonly used model for lead poisoning in animals [88,89]. Previous research [7] demonstrated that this exposure protocol resulted in blood lead levels (Pb-B) in rat offspring below the established “threshold level” of 10 µg/dL [8,9]. Therefore, the aim of this study was to maintain Pb-B levels below this threshold, and the administration of Pb-B was discontinued after weaning. At PND 28, the rat pups were anesthetized, and tissues were collected for both Pb analysis in whole blood (Pb-B) and molecular examination.

A total of 18 young animals were randomly selected from each of the control and experimental groups for further analysis. No significant differences were observed between young females (n = 18) and males (n = 18) in terms of the measured parameters (*p* = 0.5, Fisher exact test). Thus, both genders were included in the analysis. The distribution of males and females in the experimental and control groups did not differ significantly (*p* = 0.5, Fisher exact test), and there were no significant differences in weight between males (70–106 g) and females (52–85.5 g) in the two groups (*p* = 0.5, Fisher exact test).

Animals were euthanized by decapitation without anesthesia, and their brains were promptly removed, sectioned into cerebellum, hippocampus, and forebrain cortex, and stored in liquid nitrogen at −80 °C for subsequent analysis.

### 4.2. Analysis of Pb by Atomic Absorption Spectroscopy

The lead concentration in blood and brain samples was determined using inductively coupled plasma optical emission spectrometry (ICP-OES) with a Thermo Scientific ICAP 7400 Duo instrument. The ICP-OES system was equipped with a concentric nebulizer and cyclonic spray chamber to enable accurate lead measurements in both radial and axial modes.

For sample preparation, the samples were thawed to room temperature and digested using a microwave digestion system (MARS 5, CEM). Each sample, with a volume of 0.75 mL, was transferred to clean polypropylene tubes, and 2 mL of 65% nitric acid (Suprapur, Merck, Poznań, Poland) was added. The samples underwent a 30-min pre-reaction period in a clean hood. Following the pre-reaction, 1 mL of non-stabilized 30% hydrogen peroxide solution (Suprapur, Merck) was added to each vial. The samples were then placed in Teflon vessels and subjected to microwave digestion at 180 °C for 35 min (15-min ramp to 180 °C, followed by 20 min at 180 °C).

After digestion, the samples were cooled to room temperature and transferred to acid-washed 15 mL polypropylene sample tubes. A 10-fold dilution was performed prior to ICP-OES analysis. To ensure accurate measurements, the samples were spiked with an internal standard (0.5 mg/L Yttrium) and diluted to a final volume of 10 mL using a solution of 0.075% nitric acid (Suprapur, Merck) and 1 mL of 1% Triton (Triton X-100, Sigma, Poznań, Poland). The prepared samples were stored in a monitored refrigerator at 4 °C until analysis.

Blank samples, used as controls, were prepared by adding concentrated nitric acid to tubes without any sample. The blank samples were then diluted in the same manner as the test samples. Calibration standards, containing different concentrations of inorganic elements, were prepared using a similar procedure as the blanks and samples. All solutions were prepared using deionized water with a resistivity of approximately 18.0 MΩ (Direct Q UV, Millipore, Merck, Warsaw, Poland). Additionally, reference material samples (NIST SRM 8414 Bovine Muscle) were prepared following the same procedure as the test samples to validate the accuracy of the analysis. Lead measurements were conducted at a wavelength of 220.353 nm to determine the lead content in the samples.

### 4.3. Quantitative Real Time Polymerase Chain Reaction (qRT-PCR)

Quantitative analysis of mRNA expression of genes for *Iba1*, pro-inflammatory cytokines (markers of M1 microglia activation) *Il1b*, *Il6*, *Tnfa*, markers of M2 microglia activation *Arg1*, *Chi3l1*, *Mrc1*, *Il10*, *Fcgr1a*, *Tgfb1*, *Sphk1*, chemokine *Cxcl1*, *Cxcl2*, and their receptor *Cxcr2*, and *Gfap* were performed by two-step reverse transcription PCR. First, RNA extraction from the clinical specimens was performed using the RNeasy Lipid Tissue Mini Kit (Qiagen, Hilden, Germany). To eliminate DNA contamination, DNase I treatment was conducted following the manufacturer’s protocol (Sigma-Aldrich, St. Louis, MO, USA). The quantity and quality of the extracted RNA were assessed using a NanoDrop ND-1000 spectrophotometer (NanoDrop Technologies, ThermoFisher Scientific, Warsaw, Poland).

Subsequently, the first strand cDNA synthesis kit and oligo-dT primers (Fermentas, Waltham, MA, USA) were utilized for cDNA synthesis. Quantitative analysis of the target mRNAs was carried out on an ABI 7500 Fast instrument using Power SYBR Green PCR Master Mix reagent (Applied Biosystems, Waltham, MA, USA). The real time PCR reaction consisted of an initial denaturation step at 95 °C for 15 s, followed by 40 cycles of denaturation at 95 °C for 15 s, and annealing/extension at 60 °C for 1 min. The Ct values obtained were used for further analysis. Melting curve analysis confirmed the amplification of a single PCR product under the specified reaction conditions. To normalize gene expression, glyceraldehyde-3-phosphate dehydrogenase (GAPDH) was used as an endogenous control. The results were calculated as fold differences (2^dCt) and subjected to statistical analysis. The expression data were presented as absolute expression levels in the tumor tissue. The primer pairs used for amplification were as follows:
Iba1_F GATTTGCAGGGAGGAAAAGCTIba1_R AACCCCAAGTTTCTCCAGCATIl1b_F GAAATAGCACCTTTTGACAGTGIl1b_R TGGATGCTCTCATCTGGACAGIl6_F CTGCAAGAGACTTCCAGCCAGIl6_R AGTGGTATATACTGGTCTGTTGGTnfa_F CAGGCGGTGTCTGTGCCTCTnfa_R CGATCACCCCGAAGTTCAGTAGArg1_F CTCCAAGCCAAAGCCCATAGAGArg1_R GGGGCTGTCATTGGGGACATCAChi3l1_F ATGTGCACCTCTGCTGAAGCCChi3l1_R ACCAGTTTGTACGCAGAGCMrc1_F CGCTGTTCAACTCTTGGACTCMrc1_R TGGCACCCCCAAACACAATTTGAIl10_F CTTACTGGCTGGAGTGAAGACCAIl10_R TCAGCTCTCGGAGCATGTGGFcgr1a_F TCCACAAAGTGGTTTATCAACAGFcgr1a_R CACTGTCCTTGAAACTGGCCTTgfb1_F CCACCTGCAAGACCATCGACTgfb1_R CTGGCGAGCCTTAGTTTGGACSphk1_F ACTGATGCTCACCGAACGGCASphk1_R CCGTCACCGGACATGACCGCCxcl1_FCAGGGATTCACTTCAAGAACATCCxcl1_R CAGGGTCAAGGCAAGCCTCCxcl2_FCCAACCATCAGGGTACAGGCxcl2_RGGGTCACCGTCAAGCTCTGCxcr2_F TCCCTGCCCATCTTCATTCTTCCxcr2_RACCCTCCACTTGGATGTATTATGfap_FGAGTCCACAACCATCCTTCTGAGGfap_R ACACCAGGCTGCTTGAACAC

### 4.4. Measurement of Iba1, IL-1β, IL-6, TNF-α, and GFAP Concentration by ELISA Method

The measurements of Iba1, IL-1β, IL-6, TNF-α, and GFAP concentrations were conducted using appropriate immunoenzymatic ELISA Assay Kits for Iba1 (cat No. LS-F32203), TNF-α (cat No. LS-F5193), GFAP (cat No. LS-F4259), (LSBio, Seattle, WA, USA), IL-1β (cat No. ELR-IL1b-CL1), IL-6 (cat No. ELR-IL6-CL1), (Raybiotech, Poznan, Poland) according to manufacturer’s instruction.

### 4.5. Immunohistochemical Staining

Samples of dissected brains were fixed in 4% buffered formalin solution (Chempur, Piekary Śląskie, Poland) for at least 24 h and then washed with absolute ethanol (Standlab, Poland; 3 times over 3 h), absolute ethanol with xylene (Supelco, Merck, Darmstadt, Germany; 1:1; twice over 1 h) and xylene (3 times over 20 min). Then, after 3 h of saturation of the tissues in liquid paraffin, the samples were embedded in paraffin blocks. Using a microtome (Microm HM340E, Thermo Fisher Scientific, Warsaw, Poland), 3 μm serial sections were taken and placed on polysine microscope slides (Thermo Scientific, Warsaw, Poland). The sections of the brain were deparaffinized in xylene and rehydrated in decreasing concentrations of ethanol and microwaved in citrate buffer (pH 6.0) to induce epitope retrieval. After slow cooling to room temperature, slides were washed in PBS twice for 5 min and then incubated with primary antibodies overnight (4 °C). Immunohistochemistry was performed using specific primary mouse monoclonal antibody against Iba (Abcam, ab1788465, in a final dilution 1:1000) and primary rabbit monoclonal antibody against GFAP (Abcam, ab68428, in a final dilution 1:250). Sections were stained with an avidin–biotin–peroxidase system with diaminobenzidine as the chromogen (DakoEnvision+Dual Link System-HRP (DAB+)), performed according to staining procedure instructions included. Sections were washed in distilled H_2_O and counterstained with hematoxylin. The IHC reactions were performed 3 times. For a negative control, specimens were processed in the absence of primary antibodies. Positive staining was defined by visual identification of brown pigmentation using a light microscope (Leica, DM5000B, Schönwalde-Glien, Germany). The samples were independently examined by two experienced histologists.

### 4.6. Transmission Electron Microscopy (TEM) Analysis

The animals were administered Nembutal (80 mg/kg b.w.) anesthesia and perfused through the ascending aorta with a solution containing 2% paraformaldehyde and 2.5% glutaraldehyde in 0.1 M cacodylate buffer, pH 7.4 at 20 °C (Sigma-Aldrich, Poznań, Poland). Tissue samples for ultrastructural studies were collected from the CA1 region of the hippocampus, forebrain cortex, and cerebellum. The collected tissue was fixed in the same solution for 20 h and then treated with a mixture of 1% OsO_4_ and 0.8% K_4_[Fe(CN)_6_]. After dehydration using a series of ethanol and propylene oxide, the tissue specimens were embedded in Spur resin. Ultrathin sections (50 nm) were prepared and examined using a JEM 1200EX electron microscope (Jeol, Tokyo, Japan).

### 4.7. Immunochemical Determination of Protein Levels (Western Blot Analysis)

The protein level and phosphorylation status were assessed using the Western blotting method under standard conditions for immunochemical analysis. Tissue samples were homogenized and mixed with Laemmli buffer, followed by denaturation at 95 °C for 5 min. After standard SDS-PAGE separation, the proteins were transferred onto nitrocellulose membranes using wet transfer in standard conditions. The membranes were then subjected to immunochemical analysis using specific antibodies and detected using chemiluminescence.

To begin the immunochemical analysis, the membranes were washed with TBS-T (Tris-buffered saline with Tween 20 buffer: 100 mM Tris, 140 mM NaCl, and 0.1% Tween 20, pH 7.6) for 5 min to remove any non-specific binding. Subsequently, the membranes were blocked with a 3% BSA solution in TBS-T for 1 h at room temperature (RT). For GS detection, the membranes were incubated overnight at 4 °C with a primary antibody against GS (1:10,000) (G2781, Sigma-Aldrich, St. Louis, MO, USA) in 1% BSA in TBS-T. After three washes with TBS-T, the membranes were incubated with a secondary antibody (1:8000 anti-rabbit) for 60 min at RT, followed by three additional washes with TBS-T. The GS antibody was visualized using a chemiluminescent reaction and ECL reagent (Amersham Biosciences, Bath, UK) under standard conditions. As a loading control, the immunolabeling of GAPDH was performed after stripping the membranes. Densitometric analysis and verification based on size markers were carried out using TotalLab software v1.11, San Diego, CA, USA).

### 4.8. Glutamine Synthetase (GS) Activity Assay

The activity of GS was assessed in brain homogenates using a colorimetric assay based on the catalysis of γ-glutamyl hydroxamate from glutamine and hydroxylamine, following the protocols described by Bidmon et al. and Pawlik et al. [90,91]. Fragments of the hippocampus, forebrain cortex, and cerebellum were immediately frozen on dry ice after dissection and stored at −80 °C. Frozen tissue samples were then homogenized in ice-cold sucrose (0.32 M)/HEPES (2.5 mM, pH 7.5) buffer.

To measure GS activity, 50 μL portions of brain homogenates were incubated at 37 °C for 20 min with a reaction mixture containing 60 mM L-glutamine, 15 mM hydroxylamine-HCl, 20 mM Na-arsenite, 0.4 mM ADP, 3 mM MnCl_2_, and 60 mM imidazole-HCl (pH 6.8). The reaction was stopped by adding 500 μL of a stop-solution consisting of 0.2 M trichloroacetic acid, 0.67 M HCl, and 0.37 M FeCl_3_. After centrifugation (5 min at 15,000× *g*), the supernatant containing the reaction product, γ-glutamyl hydroxamate, was collected. The concentration of γ-glutamyl hydroxamate was determined colorimetrically at 500 nm using an absorbance microplate reader (SPECTROstar Nano, BMG Labtech, Ortenberg, Germany). A standard curve was generated using dilutions of γ-glutamyl hydroxamate ranging from 0.625 to 10 mM.

The protein concentration of brain homogenates was measured using the Bradford assay. GS activity was expressed as micromolar γ-glutamyl hydroxamate per minute per milligram of protein. The data were presented as a percentage (±S.E.M.) of the control value.

### 4.9. Statistical Analysis

The data are presented as mean values ± S.E.M., with each data point representing a separate animal. Normality and equality of group variances were assessed using the Shapiro–Wilk test. Differences between means were evaluated using an unpaired Student’s *t*-test and Fisher exact test for normally distributed data, while a non-parametric Mann–Whitney U test was employed for non-normally distributed data. Statistical significance was determined at *p* < 0.05. All statistical analyses were conducted using GraphPad Prism version 8.0 (GraphPad Software, San Diego, CA, USA).

## 5. Conclusions

Our findings demonstrate that low-dose pre- and neonatal exposure to Pb has a significant impact on the status of microglia and astrocytes. These cells undergo mobilization, activation, and exhibit changes in their gene expression profiles. The effects of Pb exposure on microglia are observed in a brain-structure-specific manner with distinct responses in the forebrain cortex, cerebellum, and hippocampus. In the forebrain cortex and cerebellum, both pro-inflammatory M1 microglia and potentially beneficial recovery-promoting M2 microglia are activated. However, in the hippocampus, only the classic M1 phenotype is stimulated. Notably, the forebrain cortex shows the strongest anti-inflammatory and compensatory response from microglial cells. Despite the attempt to activate the M2 phenotype of microglia, we observe a broad spectrum of pathological changes in the brain, including the forebrain cortex and cerebellum. All of this evidence suggests an inability of microglia to counteract Pb-induced pathology and achieve a state of equilibrium in the brain. Additionally, astrocyte activation, as well as disturbances in the astrocytes’ GS protein level and activity revealed an association between Pb exposure and pathological changes in astroglia. Moreover, our results suggest that the CXCL1/CXCR2 and CXCL2/CXCLR2 pathways may be involved in Pb-induced brain pathology in rat offspring during pro-inflammatory microglia and astrocyte activation. All these results suggest that both microglia and astroglia may represent a potential target for Pb toxicity, thus being key mediators of neuroinflammation and further neuropathology induced by Pb during perinatal brain development.

## Figures and Tables

**Figure 1 ijms-24-09903-f001:**
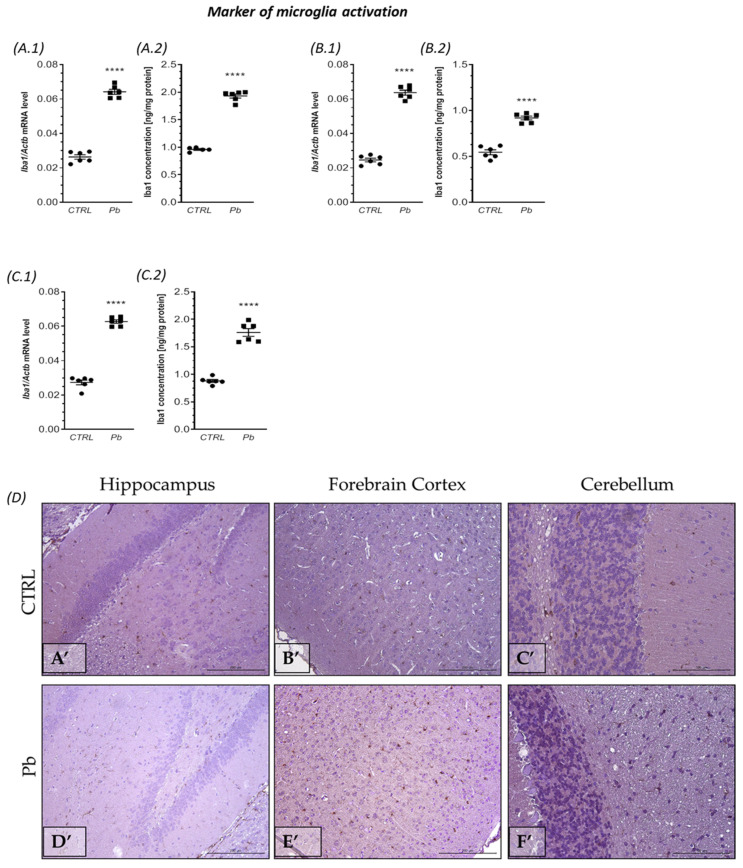
The effect of perinatal exposure to Pb on the *Iba1* mRNA expression and concentration in the brain of rat offspring. Offspring rats at PND 28 were sacrificed and brain tissues were collected. The effect of perinatal Pb exposure on the level of mRNA for *Iba1* in the hippocampus (**A.1**), forebrain cortex (**B.1**), and cerebellum (**C.1**) of offspring rats. The level of mRNA was measured by real-time PCR and calculated by the ∆∆Ct method with GAPDH as a reference gene. The effect of perinatal exposure to Pb on Iba1 concentrations in the hippocampus (**A.2**), forebrain cortex (**B.2**), and cerebellum (**C.2**). Iba1 concentrations were analyzed by enzyme-linked immunosorbent assay (ELISA) method. The effect of perinatal exposure to Pb on the immunoreactivity of Iba1 in the rat brain (**D**). Representative microphotography showing immunoexpression of Iba1 in the hippocampus, forebrain cortex, and cerebellum of control rats (**A’**–**C’**) and Pb-treated rats (**D’**–**F’**). IHC staining. Objective magnification ×20. Data represent the mean values ± SEM from n = 6 independent experiments (number of separate animals from three different litters). **** *p* < 0.0001 versus control using Student’s *t*-test.

**Figure 2 ijms-24-09903-f002:**
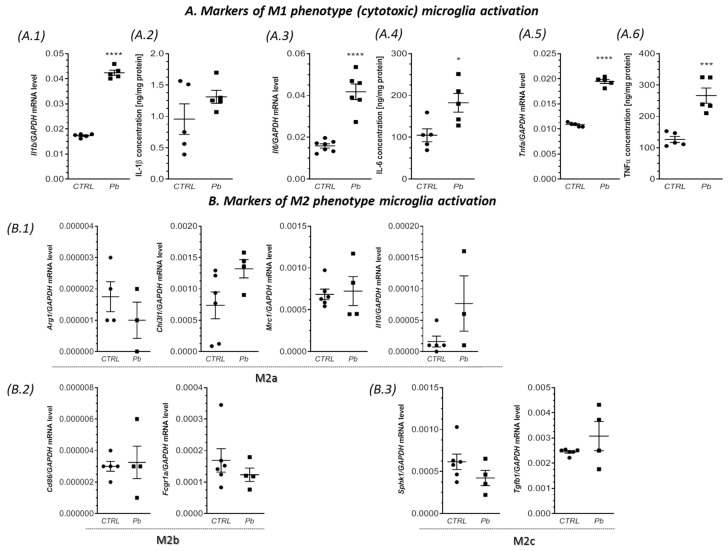
The impact of perinatal exposure to Pb on the status of microglia in the hippocampus. Offspring rats at postnatal day 28 (PND 28) were sacrificed, and their brain tissues were collected for analysis. The gene expression levels of *Il1b* (**A.1**), *Il6* (**A.3**), and *Tnfa* (**A.5**), along with the markers of M2a, M2b, and M2c subtypes’ activation including *Arg1*, *Chi3l1*, *Mrc1*, *Il10* (**B.1**), *Fcgr1a* (**B.2**), *Sphk1*, and *Tgfb1* (**B.3**) in the hippocampus of both control and Pb-exposed rats were measured using quantitative RT-PCR. The calculations were performed using the ΔΔCt method, with GAPDH as the reference gene. Additionally, the effect of perinatal Pb exposure on the concentration of IL-1β (**A.2**), IL-6 (**A.4**), and TNF-α (**A.6**) in the hippocampus was analyzed. The concentrations of IL-1β, IL-6, and TNF-α were assessed using the enzyme-linked immunosorbent assay (ELISA) method. The data presented in the study represent the mean values ± SEM from (4–6) independent experiments, including animals from three different litters. Statistical analysis was performed using Student’s *t*-test, with significance levels indicated as * *p* < 0.05, *** *p* < 0.001, **** *p* < 0.0001, compared to the control group.

**Figure 3 ijms-24-09903-f003:**
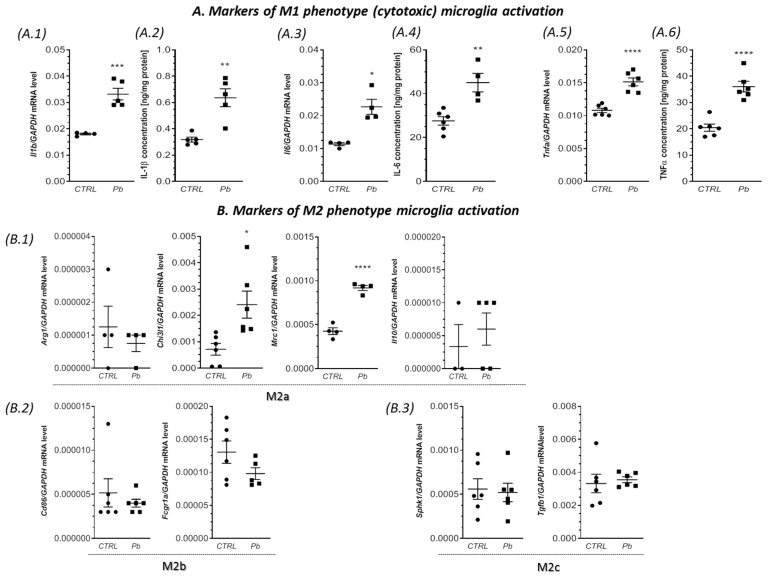
Impact of perinatal Pb exposure on microglial status in the forebrain cortex. Offspring rats were sacrificed at postnatal day 28 (PND 28), and brain tissues were collected for analysis. The gene expression levels of *Il1b* (**A.1**), *Il6* (**A.3**), and *Tnfa* (**A.5**), along with markers of M2a, M2b, and M2c subtypes’ activation, including *Arg1*, *Chi3l1*, *Mrc1*, *Il10* (**B.1**), *Fcgr1a* (**B.2**), *Sphk1*, and *Tgfb1* (**B.3**), were quantified in the forebrain cortex of both control and Pb-exposed rats. The quantification was performed using quantitative RT-PCR and calculated using the ΔΔCt method, with GAPDH serving as the reference gene. Moreover, the concentrations of IL-1β (**A.2**), IL-6 (**A.4**), and TNF-α (**A.6**) in the forebrain cortex were determined to evaluate the effect of perinatal Pb exposure. The concentrations of IL-1β, IL-6, and TNF-α were measured using the enzyme-linked immunosorbent assay (ELISA) method. The data presented in this figure represent the mean values ± SEM from (4–6) independent experiments, involving animals from three different litters. Statistical analysis was conducted using Student’s *t*-test, with significance levels indicated as * *p* < 0.05, ** *p* < 0.01, *** *p* < 0.001, and **** *p* < 0.0001, compared to the control group.

**Figure 4 ijms-24-09903-f004:**
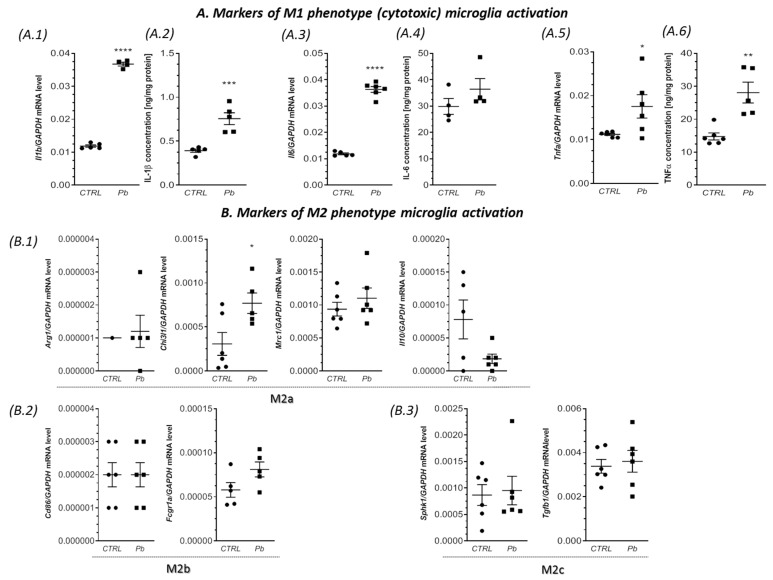
Impact of perinatal Pb exposure on microglial status in the cerebellum. Offspring rats were sacrificed at postnatal day 28 (PND 28), and brain tissues were collected for analysis. The gene expression levels of *Il1b* (**A.1**), *Il6* (**A.3**), and *Tnfa* (**A.5**), along with markers of M2a, M2b, and M2c subtypes’ activation, including *Arg1*, *Chi3l1*, *Mrc1*, *Il10* (**B.1**), *Fcgr1a* (**B.2**), *Sphk1*, and *Tgfb1* (**B.3**), were quantified in the cerebellum of both control and Pb-exposed rats. The quantification was performed using quantitative RT-PCR and calculated using the ΔΔCt method, with GAPDH serving as the reference gene. Additionally, the concentrations of IL-1β (**A.2**), IL-6 (**A.4**), and TNF-α (**A.6**) in the cerebellum were determined to evaluate the effect of perinatal Pb exposure. The concentrations of IL-1β, IL-6, and TNF-α were measured using the enzyme-linked immunosorbent assay (ELISA) method. The data presented in this figure represent the mean values ± SEM from (4–6) independent experiments, involving animals from three different litters. Statistical analysis was conducted using Student’s *t*-test, with significance levels indicated as * *p* < 0.05, ** *p* < 0.01, *** *p* < 0.001, and **** *p* < 0.0001, compared to the control group.

**Figure 5 ijms-24-09903-f005:**
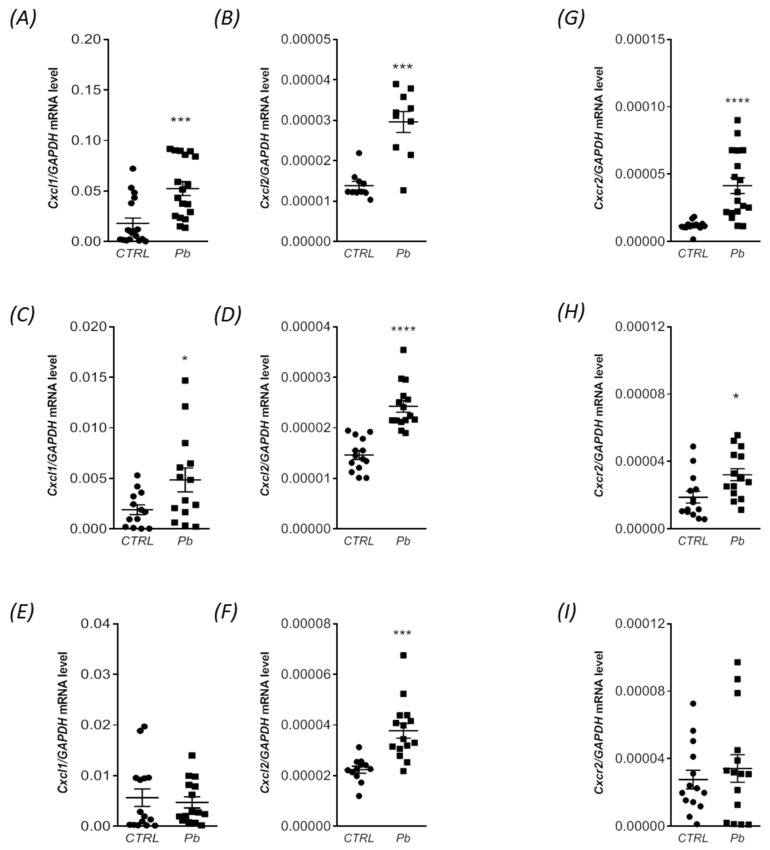
Impact of perinatal Pb exposure on gene expression of *Cxcl1*, *Cxcl2*, and *Cxcr2* in rat offspring brain. Offspring rats were sacrificed at postnatal day 28 (PND 28), and brain tissues were collected for analysis. The effect of perinatal Pb exposure on the levels of mRNA for *Cxcl1*, *Cxcl2*, and *Cxcr2* in the hippocampus (**A**,**B**,**G**, respectively), forebrain cortex (**C**,**D**,**H**, respectively), and cerebellum (**E**,**F**,**I**, respectively) of the offspring rats was examined. The quantification of mRNA levels was performed using quantitative RT-PCR and calculated using the ∆∆Ct method, with GAPDH serving as the reference gene. The presented data represent the mean values ± SEM from six independent experiments, encompassing animals from three different litters. Statistical analysis was conducted using Student’s *t*-test, with significance levels indicated as * *p* < 0.05, *** *p* < 0.001, and **** *p* < 0.0001 compared to the control group.

**Figure 6 ijms-24-09903-f006:**
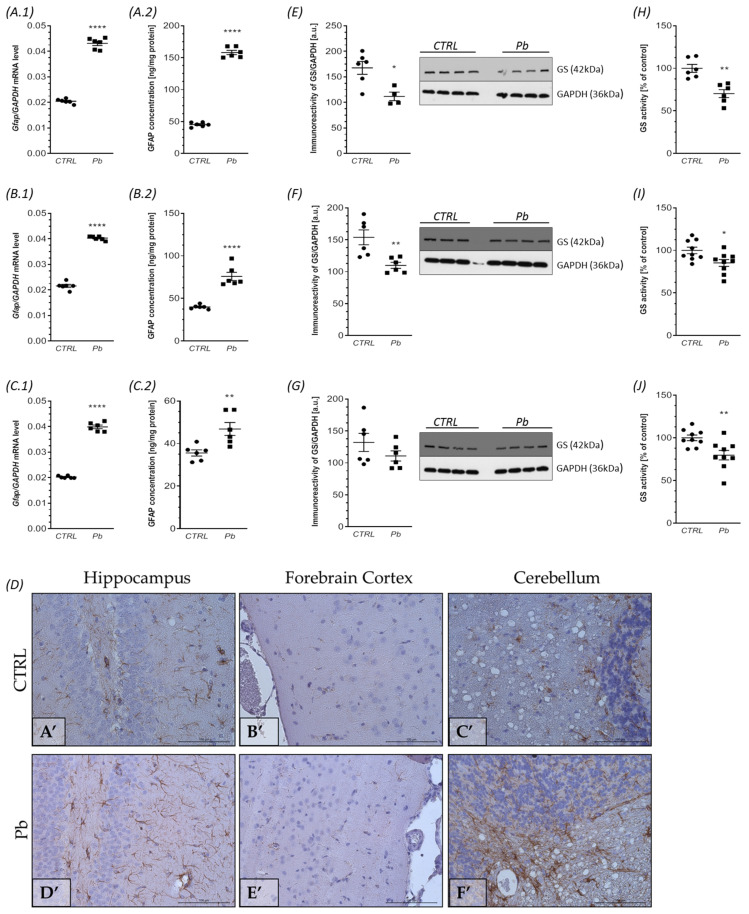
The effect of perinatal exposure to Pb on the astrocyte phenotype and function in the brain of rat offspring. Offspring rats at postnatal day 28 (PND 28) were sacrificed and brain tissues were collected. The effect of perinatal Pb exposure on levels of mRNA for *Gfap* in the hippocampus (**A.1**), forebrain cortex (**B.1**), and cerebellum (**C.1**) of rat offspring. The level of mRNA was measured by quantitative RT-PCR and calculated by the ∆∆Ct method with GAPDH as a reference gene. The effect of perinatal exposure to Pb on GFAP concentrations in the hippocampus (**A.2**), forebrain cortex (**B.2**), and cerebellum (**C.2**). Concentrations of GFAP were analyzed by enzyme-linked immunosorbent assay (ELISA) method. The effect of perinatal exposure to Pb on the immunoreactivity of GFAP in the rat brain (**D**). Representative microphotography showing immunoexpression of GFAP (brown color reaction in the hippocampus, forebrain cortex and cerebellum of control rats (**A’**–**C’**) and Pb-treated rats (**D’**–**F’**). IHC staining. Objective magnification ×40. The immunoreactivity of the GS protein in the control and Pb-exposed rats was monitored using a Western blot analysis. Densitometric analysis and representative pictures of GS in the hippocampus (**E**), forebrain cortex (**F**), and cerebellum (**G**) are shown. Results were normalized to GAPDH levels. Data represent the mean values ± SEM from n = 6 independent experiments (number of separate animals from three different litters). The effect of perinatal exposure to Pb on the activity of GS in the hippocampus (**H**), forebrain cortex (**I**), and cerebellum (**J**). The GS activity was measured with a colorimetric assay, as described in the Methods section. Data represent the mean values ± SEM from n = (6–9) independent experiments (number of separate animals from three different litters). * *p* < 0.05, ** *p* < 0.01, **** *p* < 0.0001 versus control using Student’s *t*-test.

**Figure 7 ijms-24-09903-f007:**
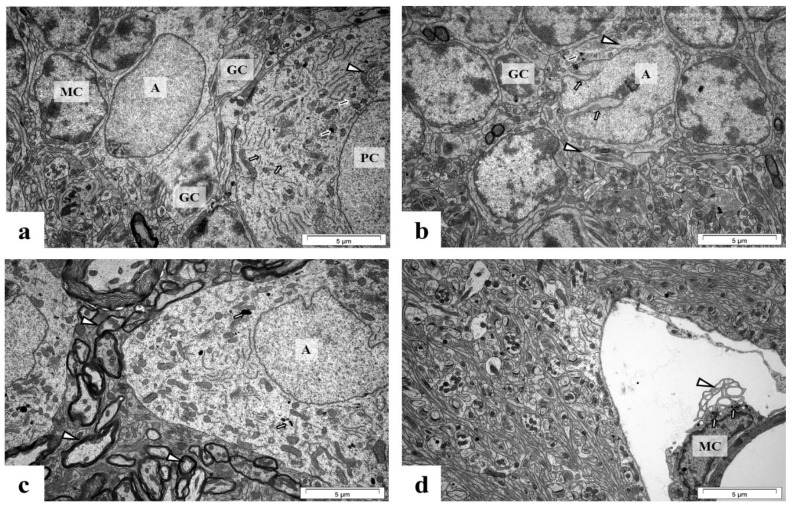
Cerebellum from control (**a**) and perinatal Pb exposure rats (**b**–**d**). In the control rat’ cerebellum (**a**) are visible Purkinje cells (PC) with fragmented smooth ER (white arrow), numerous lysosomes (black arrow), and active Golgi apparatus (arrowhead) in the cytoplasm, near normal astrocyte (A), normal granule cells (GC) and normal inactive microglia cell (MC). In cerebellum from perinatal Pb exposure rats (**b**) there is visible active astrocyte (A) with nuclear protrusions (white arrow), single lysosomes (black arrow), and GFAP (arrowhead) in the cytoplasm, near normal granule cells (GC); (**c**) Visible altered, stratified myelin sheath (arrowheads), active astrocyte (A) with single lysosomes (white arrows) in the cytoplasm; (**d**) microglia cell (MC) with cytoplasmic lysosomes (white arrow) and pinocytotic vacuoles (arrowhead). Scale bar: 5 μm.

**Figure 8 ijms-24-09903-f008:**
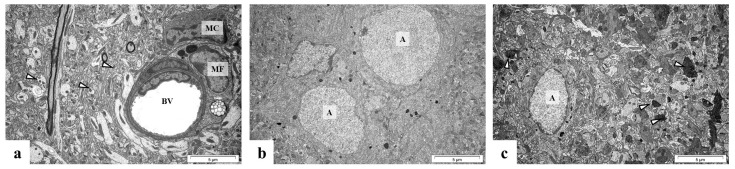
Forebrain cortex from control (**a**) and perinatal Pb exposed rats (**b**,**c**). In the control forebrain cortex (**a**) there are present capillary blood vessels (BV) with slightly folded endothelium and perivascular macrophage (MF), near microglial cell (MC), and unchanged neuropil (arrowheads). After perinatal Pb exposure, in the forebrain cortex active astrocytes (A) are visible (**b**,**c**) and the mass accumulation of vesicles (arrowheads) in the presynaptic part of synapses draws attention here (**c**). Scale bar: 5 μm.

**Figure 9 ijms-24-09903-f009:**
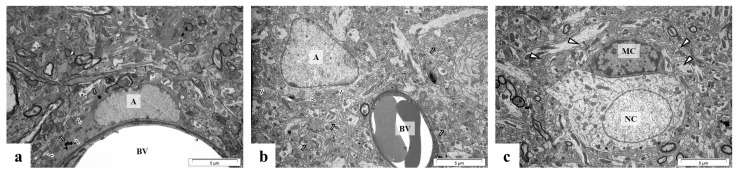
CA1 region of hippocampus from control rat (**a**) and and rat intoxicated with lead (**b**,**c**). In control hippocampus (**a**) there are vissible precapillary vessel (BV), near astrocyte (A) with GFAP (black arrows) and lysosomes (white arrows) within cytoplasm. After perinatal Pb exposure, in hippocampus are visible: (**b**) capillary vessel (BV) with erythrocytes tightly adhering to the wall of the vessel, near active astrocyte (A) with GFAP (black arrows) in the cytoplasm and ultrastructurally unchanged neuropil (white arrows); (**c**) nerve cell (NC), near microglial cell (MC) with pinocytotic vacuoles (arrowheads) indicating its activity. Scale bar: 5 μm.

**Figure 10 ijms-24-09903-f010:**
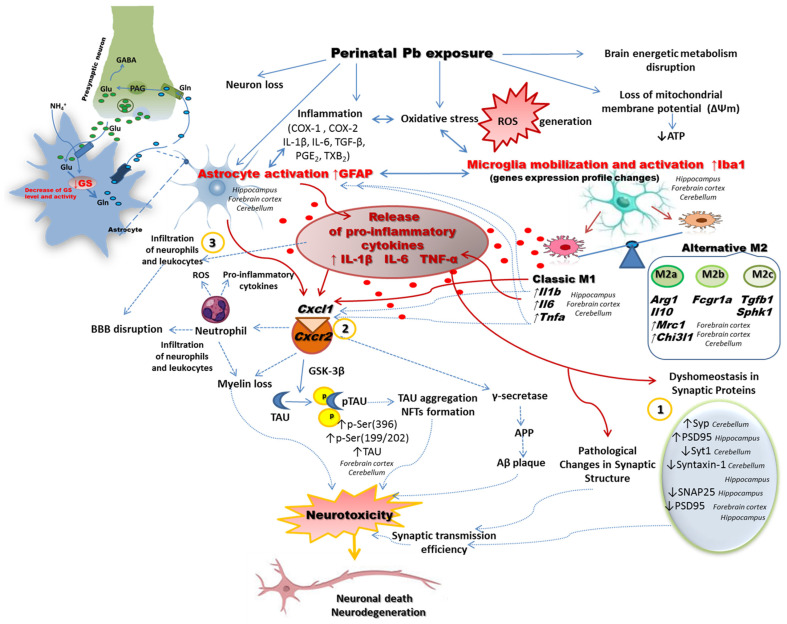
The schematic diagram illustrates the detrimental effects of prenatal and neonatal Pb exposure on the developing brain of rat offspring. Our study identified various pathological processes that are triggered by perinatal Pb exposure, including inflammation, generation of reactive oxygen species (ROS) with oxidative stress, and disruption of brain energy metabolism. These processes contribute to neuroinflammation, activation, and mobilization of microglia into the M1 phenotype, as well as astrogliosis. It is proposed that these events set off a molecular cascade leading to several consequences: (1) synaptopathology, characterized by brain-structure-specific abnormalities in the expression of crucial pre- and postsynaptic proteins, ultimately resulting in pathological changes in synaptic structure; (2) upregulation of chemokines, such as CXCL1; and (3) impairment of the blood–brain barrier (BBB) and infiltration of neutrophils and leukocytes. Our study findings demonstrate that Pb exposure has enduring effects on the status of microglia, with brain-region-specific responses. In the forebrain cortex and cerebellum, both the pro-inflammatory M1 phenotype and the potentially beneficial recovery-promoting M2 phenotype were activated. However, in the hippocampus, only the classic M1 phenotype was stimulated. Microglia responded to Pb exposure by releasing various cytotoxic substances, including pro-inflammatory cytokines (IL-1β, IL-6, TNF-α), reactive oxygen species (ROS), and chemokines (such as CXCL1), which contributed to further neuronal damage in the brain. IL-1β, released by activated microglia, can stimulate the secretion of other cytokines, particularly IL-6, from astrocytes, promoting an inflammatory environment. TNF-α, another pro-inflammatory cytokine released by microglia, can activate astrocytes, leading to the production of CXCL1, a chemokine crucial for the recruitment of neutrophils. CXCL1, through its receptor CXCR2, can induce hyperphosphorylation of the Tau protein, initiating the formation of neurofibrillary tangles (NFTs), a hallmark of neurodegenerative diseases. Additionally, CXCL1 may enhance the activity of γ-secretase, resulting in an increased release of amyloid-beta (Aβ) and the formation of amyloid plaques. Furthermore, CXCL1 can attract neutrophils, which generate ROS, pro-inflammatory cytokines, and secrete elastase, an enzyme that disrupts the blood–brain barrier (BBB). Activation of CXCR2 inhibits the differentiation and migration of oligodendrocyte precursor cells, impairing remyelination and leading to myelin loss. Moreover, Pb exposure triggers astrocyte activation, leading to the secretion of various inflammatory cytokines and chemokines, including CXCL1. These inflammatory mediators further activate microglia and macrophages and contribute to detrimental reactions that can disrupt the integrity of BBB tight junctions. Overall, the neurotoxic effects of Pb exposure involve the activation of microglia and astrocytes, the release of inflammatory mediators, and the subsequent cascade of immune responses, ultimately leading to neuronal injury, Tau pathology, amyloid plaque formation, BBB disruption, and myelin loss. Moreover, the present study demonstrated that Pb administration caused a decrease in the protein level and activity of glutamine synthetase (GS), leading to glutamate (Glu) and glutamine (Gln) dyshomoeostasis. Depletion of Gln, a precursor for neurotransmitters glutamate and GABA, disrupts synaptic transmission and plasticity. Additionally, a decrease in GS activity mediates ammonia toxicity. All these processes have a neurotoxic effect on nerve cells, leading to the degradation of neurons underlying the etiology of neurodegenerative and neurodevelopmental diseases. Glial cell activation following perinatal Pb exposure may contribute to the initiation of processes that lead to the death of neurons. All these results suggest that neuroglia cells (microglia and astrocytes) may represent a potential target for the manipulation of Pb-induced neuroinflammatory injury of the brain. Microglia and astrocytes may play a crucial role as mediators of the inflammatory process during perinatal brain injury induced by Pb. (Arrow)—Processes and changes evoked by Pb exposure such as inflammation, oxidative stress with ROS generation, brain energetic metabolism disruption, Tau pathology, and pathological changes in the synaptic structure associated with disturbed expression of key synaptic proteins, which have been demonstrated in our previous studies. In the current report, we observed reactive astrogliosis and microglial activation, which were associated with the pro-inflammatory cytokine release, an increase in the expression of CXCL1 and CXCR2, and a decrease in the level and activity of GS. (Dashed arrow)—Proposed processes and molecular mechanisms of Pb action.

## Data Availability

The raw data supporting the conclusions of this article will be made available by the authors without undue reservation.

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
