# Peer review of "Microglia and Astroglia—The Potential Role in Neuroinflammation Induced by Pre- and Neonatal Exposure to Lead (Pb)"

_ijms, 2023, doi:10.3390/ijms24129903_

Round 1

Reviewer 1 Report

In the present study, the authors aimed to investigate the involvement of glial cells in the neuroinflammation induced by exposure to Pb using a rat model that mimics the effects of low-dose environmental Pb exposure with blood Pb levels below the threshold previously considered as safe. The authors determined mRNA and protein levels of glial markers (Iba-1 and GFAP) as well as inflammatory markers (mRNA of several pro-inflammatory and anti-inflammatory cytokines and chemokines) in several brain regions (hippocampus, forebrain cortex and cerebellum). They also examined the ultrastructural changes in the brain of rats exposed to Pb.

 The main conclusion of the manuscript is that perinatal exposure to PB at low doses affects both microglial and astrocyte cells in a brain-structure-specific manner.

This study is largely descriptive and well executed. The paper is also well written and organized. Although some results, as the increase in the pro-inflammatory cytokine levels, have already been described previously (Chibowska et al., 2020), the authors advance the current knowledge in the field of glial cell involvement in the neuroinflammatory response induced by perinatal Pb exposure. However, the authors should consider addressing several major points prior to publication.

Major comments

1.    The introduction section is too long. I recommend modifying this section to describe briefly the current state of the research field. I consider important to briefly mention the main aim of the work and highlight the main conclusions.

2.    In my opinion, this paper could be further improved by adding immunohistochemical experiments. Therfore, immunohistochemical images showing the expression of Iba-1 and GFAP in the brain regions analyzed should be presented. Even, the authors could also consider an additional IHC marker to confirm microglial activation, as MHCII, a relevant marker of activated microglia.

3.    Although the majority of the markers of M2 phenotype microglia activation are unaltered, a significant increase in the mRNA levels of Chi3l1 and Mrc1 is observed in the forebrain cortex and cerebellum. It is recommended that the authors add protein levels (Western blot or ELISA) of these cytokines to support the data of the qPCR and their general conclusions.

4.    A major concern is the over-interpretation of the data presented. In particular, the authors declare several times in introduction and discussion that the Pb exposure exacerbate CNS damage promoting glial activation, but is there any cell death of neurons in this model? The results of this study demonstrate ultrastructural changes, but there is no data supporting CNS damage induced by Pb exposure. Estimation of number of neurons (neuN or nissl stain) might be relevant.

5.    It is concluded that perinatal exposure to Pb at low doses affects glial cell status in a brain-structure-specific manner. The authors provide a weak discussion on why neuroinflammatory response is brain-structure-specific. Moreover, in lines 503-507 it is affirmed that “It is plausible that to reestablish homeostasis in the brain, microglial cells in the forebrain cortex and cerebellum may have switched from a pro-inflammatory M1 phenotype to a neuroprotective M2 phenotype during the progression of Pb-induced pathology. This response may have been due to the wider range of pathological changes observed in the forebrain cortex compared to those in the hippocampus or cerebellum” what pathological changes are you referring to? The data presented does not support this conclusion, in fact, TEM analysis of the rat brains show ultrastructural changes mainly in cerebellum, while forebrain cortex and hippocampus seem to be less ultrastructurally sensitive. In addition, previous data demonstrated ultrastructural alterations in hippocampal neurons (Baranowska-Bosiacka et al., 2013). The authors need to discuss better this point.

6.    In the discussion, the authors state that “Pb exposure exacerbates CNS damage by promoting M1 polarization and inhibiting M2 polarization of hippocampal microglia in rat brains” (lines 495-497). There is no data in this manuscript supporting an inhibition of M2 polarization state.

Minor comments

-          Take care of the abbreviations, once defined you should use the abbreviated form. In particular, Pb or lead is used indistinctly.

Author Response

Dear Editor,

Dear Reviewers,

First of all, we would like to thank you very much for consideration of our manuscript: “Microglia and Astroglia – The Potential Role in Neuroinflammation Induced by Pre- and Neonatal Exposure to Lead (Pb)” for publications in the IJMS (Manuscript ID: IJMS-2383916). We have taken into consideration very carefully all remarks and comments of all  Reviewers. The manuscript was corrected according to the Reviewer’s remarks (all changes are marked in red colour in review).

The major changes suggested by Reviewers:

  • According to Reviewer suggestion the Introduction was shortened and more focused on the main issue of work. The main goal and conclusions were highlighted. All made changes are highlighted in red.
  • According to Reviewer suggestion the manuscript was enriched with additional immunohistochemical experiments showing the expression of Iba1 and GFAP in all analysed brain structures (Figure 1D and Figure 6D in revised manuscript).
  • According to Reviewer suggestion Discussion was corrected.
  • The manuscript was linguistically revised by a native English-speaking colleague.

We consider that all corrections increased the value of our article and our explanations you will find adequate. I hope our corrected manuscript can be accepted for publication in IJMS.

I would very much appreciate your opinion thereon.

Yours faithfully,

Irena Baranowska-Bosiacka

Answers from the authors of the manuscript to Reviewers’ questions:

Answer to Reviewer 1

Reviewer #1: In the present study, the authors aimed to investigate the involvement of glial cells in the neuroinflammation induced by exposure to Pb using a rat model that mimics the effects of low-dose environmental Pb exposure with blood Pb levels below the threshold previously considered as safe. The authors determined mRNA and protein levels of glial markers (Iba-1 and GFAP) as well as inflammatory markers (mRNA of several pro-inflammatory and anti-inflammatory cytokines and chemokines) in several brain regions (hippocampus, forebrain cortex and cerebellum). They also examined the ultrastructural changes in the brain of rats exposed to Pb. The main conclusion of the manuscript is that perinatal exposure to PB at low doses affects both microglial and astrocyte cells in a brain-structure-specific manner. This study is largely descriptive and well executed. The paper is also well written and organized. Although some results, as the increase in the pro-inflammatory cytokine levels, have already been described previously (Chibowska et al., 2020), the authors advance the current knowledge in the field of glial cell involvement in the neuroinflammatory response induced by perinatal Pb exposure. However, the authors should consider addressing several major points prior to publication.

Major comments:

  1. The introduction section is too long. I recommend modifying this section to describe briefly the current state of the research field. I consider important to briefly mention the main aim of the work and highlight the main conclusions.
  2. In my opinion, this paper could be further improved by adding immunohistochemical experiments. Therefore, immunohistochemical images showing the expression of Iba-1 and GFAP in the brain regions analysed should be presented. Even, the authors could also consider an additional IHC marker to confirm microglial activation, as MHCII, a relevant marker of activated microglia.
  3. Although the majority of the markers of M2 phenotype microglia activation are unaltered, a significant increase in the mRNA levels of Chi3l1 and Mrc1 is observed in the forebrain cortex and cerebellum. It is recommended that the authors add protein levels (Western blot or ELISA) of these cytokines to support the data of the qPCR and their general conclusions.
  4. A major concern is the over-interpretation of the data presented. In particular, the authors declare several times in introduction and discussion that the Pb exposure exacerbate CNS damage promoting glial activation, but is there any cell death of neurons in this model? The results of this study demonstrate ultrastructural changes, but there is no data supporting CNS damage induced by Pb exposure. Estimation of number of neurons (neuN or nissl stain) might be relevant.
  5. It is concluded that perinatal exposure to Pb at low doses affects glial cell status in a brain-structure-specific manner. The authors provide a weak discussion on why neuroinflammatory response is brain-structure-specific. Moreover, in lines 503-507 it is affirmed that “It is plausible that to reestablish homeostasis in the brain, microglial cells in the forebrain cortex and cerebellum may have switched from a pro-inflammatory M1 phenotype to a neuroprotective M2 phenotype during the progression of Pb-induced pathology. This response may have been due to the wider range of pathological changes observed in the forebrain cortex compared to those in the hippocampus or cerebellum” what pathological changes are you referring to? The data presented does not support this conclusion, in fact, TEM analysis of the rat brains show ultrastructural changes mainly in cerebellum, while forebrain cortex and hippocampus seem to be less ultrastructurally sensitive. In addition, previous data demonstrated ultrastructural alterations in hippocampal neurons (Baranowska-Bosiacka et al., 2013). The authors need to discuss better this point.
  6. In the discussion, the authors state that “Pb exposure exacerbates CNS damage by promoting M1 polarization and inhibiting M2 polarization of hippocampal microglia in rat brains” (lines 495-497). There is no data in this manuscript supporting an inhibition of M2 polarization state.

Minor comments

-          Take care of the abbreviations, once defined you should use the abbreviated form. In particular, Pb or lead is used indistinctly.

Answer: We are very grateful for the examination of our manuscript and for valuable comments, and also for time offered for the preparation of revision. We fully agree you’re your opinion. We have carefully taken into consideration all remarks and comments. The manuscript was improved according to your suggestions. All changes to the manuscript are highlighted. We believe that these changes have resulted in a substantially strengthened manuscript for potential publication in IJMS. Please find below our response.

Point 1 The introduction section is too long. I recommend modifying this section to describe briefly the current state of the research field. I consider important to briefly mention the main aim of the work and highlight the main conclusions.

Answer: Thank you very much for this remark. We fully agree with your opinion. The Introduction section has been shortened and modified according to your suggestion. The main goal and conclusions were highlighted.

Point 2 In my opinion, this paper could be further improved by adding immunohistochemical experiments. Therefore, immunohistochemical images showing the expression of Iba-1 and GFAP in the brain regions analysed should be presented. Even, the authors could also consider an additional IHC marker to confirm microglial activation, as MHCII, a relevant marker of activated microglia.

Answer: Thank you very much for these remarks. We fully agree with your opinion. According to this, we performed immunohistochemical tests showing the expression of Iba1 and GFAP in the studied brain regions. Our ELISA results were confirmed by immunohistochemical analysis, which revealed significantly higher immunoexpression of Iba1 in all examined brain structures of Pb-treated rats (Figure 1D, D’-F’) than in control animals (Figure 1D, A’-C’). In the control (Figure 1D, A’) and Pb-treated (Figure 1D, D’) rats the Iba1-positive cells (microglia cells, brown-coloured cells) were present in the neuropil of the hippocampus between nerve cells of Gyrus Dentate (GD) and Cornu Ammonis (both structures visible), and Iba1-immunopositive cells were much more numerous after Pb-intoxication (Figure 1D, D’). In the neuropil of the neocortex of control rats (Figure 1D, B’), the Iba-reactive cells were occasionally visible in contrast to Pb-treated rats (Figure 1D, E’), in which the Iba-immunopositive cells were more prominent in all six layers of the gray matter. In the cerebellum, Iba-positive cells were mainly in the Granular Cell Layer of gray matter and in the white matter, however, in Pb-treated rats the immunoexpression of Iba in these areas of the cerebellum was significantly higher (much more microglia cells were visible, compared to control) (Figure 1D, C’ and F’).

Figure 1D in the manuscript after major revision. The effect of perinatal exposure to Pb on the immunoreactivity of Iba1 in the rat brain (D). Representative microphotography showing immunoexpression of Iba1 in the hippocampus, neocortex and cerebellum of control rats (A’-C’) and Pb-treated rats (D’-F’). IHC staining. Objective magnification x20.

Also immunohistochemical analysis of GFAP confirmed our results obtained using the ELISA method. After Pb intoxication the number of astrocytes (GFAP-positive cells) were significantly increased (mainly in the hippocampus and cerebellum) (Figure 6D, D’-F’) in comparison to control animals (Figure 6D, A’-C’). In the control and Pb-treated rats the GFAP-positive cells (astrocytes, brown-coloured cells) were present in the neuropil of the hippocampus between nerve cells of Gyrus Dentate (GD) and Cornu Ammonis (not shown on microphotography), and GFAP-immunopositive cells were much more numerous after Pb-intoxication (Figure 6D, A’ and D’). In the neuropil of the neocortex of control and Pb-treated rats, the GFAP-reactive cells were occasionally visible but were more prominent in Pb-treated rats (Figure 6D, B’ and E’). In the cerebellum, GFAP-positive cells were mainly observed in the Granular Cell Layer of gray matter and in the white matter, and in these areas of the Pb-treated rats’ cerebellum there were much more astrocytes (Figure 6D, C’ and F’).

Figure 6D in the manuscript after major revision. The effect of perinatal exposure to Pb on the immunoreactivity of GFAP in the rat brain. Representative microphotography showing immunoexpression of GFAP (brown colour reaction in the hippocampus, neocortex and cerebellum of control rats (A’-C’) and Pb-treated rats (D’-F’). IHC staining. Objective magnification x40.

Very important for our research is also the comment of the reviewer, which indicates the need to conduct an analysis that would allow distinguishing between microglia and monocytes/macrophages in the brain using the MHCII marker. This is a particularly interesting issue in terms of damage to the blood-brain barrier in Pb poisoning. Unfortunately, at the moment we cannot perform this analysis due to the lack of adequate funds for research using new antibodies or microarray techniques. But it is a very important direction for our future research.

Point 3 Although the majority of the markers of M2 phenotype microglia activation are unaltered, a significant increase in the mRNA levels of Chi3l1 and Mrc1 is observed in the forebrain cortex and cerebellum. It is recommended that the authors add protein levels (Western blot or ELISA) of these cytokines to support the data of the qPCR and their general conclusions.

Answer: Thank you very much for these remarks. We fully agree with your opinion. Unfortunately, we were unable to perform the Western blot for Mrc1 and CHI3L for reasons beyond our control (the need to order new antibodies and a long waiting time/the antibodies that we had didn't work).

Point 4 A major concern is the over-interpretation of the data presented. In particular, the authors declare several times in introduction and discussion that the Pb exposure exacerbate CNS damage promoting glial activation, but is there any cell death of neurons in this model? The results of this study demonstrate ultrastructural changes, but there is no data supporting CNS damage induced by Pb exposure. Estimation of number of neurons (neuN or nissl stain) might be relevant.

Answer: Thank you very much for this remark. Studies investigating whether pre- and neonatal exposure to Pb results in apoptosis of neurons were investigated in detail in our previous work [Baranowska-Bosiacka I. et al. Perinatal exposure to lead induces morphological, ultrastructural and molecular alterations in the hippocampus. Toxicology 2013, 303:187- 200]. In this study, we examined the activity and expression of the caspase-3 gene, translocation of AIF into the cell nucleus, DNA fragmentation, expression of Bax and Bcl-2 genes and proteins, as well as the concentration of BDNF in the forebrain cortex, cerebellum and hippocampus. We showed that exposure to Pb had no effect on the activity of caspase-3 in any of the examined brain structures. We also found a decrease in the expression of Bax mRNA and protein in the hippocampus and forebrain cortex, and a lower expression of Bcl-2 mRNA and protein in all examined brain regions of rats exposed to Pb compared to the control group. The Bax/Bcl-2 ratio (both at the level of gene and protein expression) was lower in the cerebral cortex and hippocampus of rats exposed to Pb than in the control group. Our study showed that pre- and neonatal exposure of rats to Pb resulted also in a decrease in the expression of AIF mRNA in studied regions of the rat brain and did not cause the translocation of the AIF protein from the mitochondria to the cell nucleus. In the available literature, no papers analyzing changes in mRNA or AIF protein expression under Pb toxicity were found. It seems likely that the observed decrease in Bax expression could be the reason for the observed lack of nuclear expression of the AIF protein. In the conducted TUNEL staining studies, we showed the presence of only a few apoptotic neurons in the hippocampus of control rats. The apoptotic cell index in the control group was 1.13% in the Granule Cell Layer of the Dentate Gyrus and 2.76-4.58% in the horn of Ammon of the hippocampus. In contrast, the index of apoptotic cells in the hippocampus of rats exposed to Pb was 1.54-4.95%. In this study, on the basis of electrophoresis and propidium iodide staining, we did not detect DNA degradation in any of the examined brain areas in Pb-exposed rats or in controls. The data obtained in our own research indicate that exposure to Pb during the developmental period resulting in Pb-B below 10 ug/dL did not cause the appearance of signs of apoptosis. The quantitative advantage of the anti-apoptotic Bcl-2 protein over Bax could inhibit apoptosis and could be responsible for the lack of caspase activation and the lack of presence of the AIF protein in the nucleus found in the brain of rats exposed to Pb during the pre- and neonatal period. However, what is worth highlighting, in our study we showed that the number of pyramidal neurons in the CA1 region and the thickness of the Pyramidal Cell Layer of the hippocampus were significantly lower in rats exposed to Pb compared to the control group. Similarly, the thickness of the Granular Layer of the Dentate Gyrus and the number of granular neurons in the hippocampus of Pb-exposed rats were significantly lower than in controls. The obtained results indicate that exposure to Pb during development may reduce the number of both pyramidal cells in the CA1 region of the hippocampus and granular neurons of the Dentate Gyrus.

In this study, a reduced concentration of BDNF was also demonstrated in the brains of rats exposed to Pb in relation to the control group. Since the formation and release of BDNF are influenced by the activation of postsynaptic NMDA receptors, it is possible that the inhibition of the receptor activity and the disturbance of its expression under the influence of Pb was the cause of the decrease in BDNF concentration in the brains of rats exposed to Pb, observed in our studies. Therefore, it seems that the reduction in the concentration of trophic factors necessary for the survival of neurons can be indicated as one of the possible causes of the observed decrease in the number of neurons in the presented own research. An alternative cause of the reduction in the number of neurons, related to exposure to Pb during fetal life and maternal feeding, could be apoptosis or programmed neuronal necrosis. However, our own research did not provide evidence of apoptosis of neurons as the cause of their reduced number. However, since Pb was administered prenatally, it cannot be ruled out that the apoptosis process took place in this strategic period for brain development. Impairment of neuronal differentiation under the influence of Pb seems more likely.

Point 5 It is concluded that perinatal exposure to Pb at low doses affects glial cell status in a brain-structure-specific manner. The authors provide a weak discussion on why neuroinflammatory response is brain-structure-specific. Moreover, in lines 503-507 it is affirmed that “It is plausible that to reestablish homeostasis in the brain, microglial cells in the forebrain cortex and cerebellum may have switched from a pro-inflammatory M1 phenotype to a neuroprotective M2 phenotype during the progression of Pb-induced pathology. This response may have been due to the wider range of pathological changes observed in the forebrain cortex compared to those in the hippocampus or cerebellum” what pathological changes are you referring to? The data presented does not support this conclusion, in fact, TEM analysis of the rat brains show ultrastructural changes mainly in cerebellum, while forebrain cortex and hippocampus seem to be less ultrastructurally sensitive. In addition, previous data demonstrated ultrastructural alterations in hippocampal neurons (Baranowska-Bosiacka et al., 2013). The authors need to discuss better this point.

Answer: Thank you very much for your in-depth and inspiring review of our Manuscript and for all your comments. We fully agree with Reviewer's suggestions and have deeply discussed why the neuroinflammatory response is brain-structure-specific. The Discussion section has been modified and enriched with new explanations, taking into account the issues raised by the Reviewer.

“The broad spectrum of pathological changes induced by perinatal Pb exposure in the rat brain, as indicated in our previous studies (e.g. oxidative stress with reactive oxygen species (ROS) generation, impairment of pro- and antioxidative balance of neurons, ultrastructural and molecular alterations in synapses, Tau protein pathology, disturbed brain energy metabolism, and neurons loss) [6,7,65], may be responsible for the strong and almost exclusively pro-inflammatory response of microglia, particularly in the hippocampus, where only an cytotoxic M1 phenotype was stimulated. The gene expression of all analysed markers of the M2 phenotype (M2a, M2b, and M2c subtypes): Arg1, Chi3l1, Mrc1, Fcgr1a, Tgfb1, and Sphk1, was unaffected in the hippocampus of the offspring exposed perinatally to Pb. The lack of changes in gene expression of the M2 phenotype may reflect the inability of this brain structure to restore homeostasis in response to wide-scope pathological changes and abnormalities evoked by exposure to Pb. Changes in the microglia profile are correlated with the type of challenge faced by the CNS. Taking into account the multi-directional neurotoxicity action of Pb in the hippocampus (where oxidative stress and impairment of pro- and antioxidative balance of neurons, as well as a decrease in a number of neurons concomitantly with ultrastructural pathological alterations, were the most strongly observed and revealed by us in our previous studies) [4-7], any attempt to change microglia phenotype to one that will allow for the repair and reconstruction of damaged tissue in this brain structure turns out to be ineffective and downright impossible. All these results may suggest also a greater sensitivity of the hippocampus to Pb-induced pathology unlike the forebrain cortex or cerebellum. Long-term overactivation of microglia releases various harmful factors which exert stress on neurons, causing gradual loss of neuronal function or leading to their death. In the hippocampus, we observed the strongest loss of neurons (the number of pyramidal neurons in the CA1 region and granular neurons as well as the thickness of both the Granular Layer of the Dentate Gyrus and Pyramidal Cell Layer of the hippocampus were significantly lower in rats exposed to Pb compared to the control group) [5].

Our findings suggest that Pb exposure has the potential to worsen CNS damage by promoting M1 polarization of microglia in the hippocampus of rat brains. Additionally, in the forebrain cortex and cerebellum, where pro-inflammatory cytokine expression was significantly altered, perinatal Pb exposure also affected the expression of key markers associated with alternative, anti-inflammatory, and neuroprotective microglia (M2 phenotype). Notably, in a brain structure-specific manner, there were varying degrees of activation of the potentially beneficial recovery-promoting microglia phenotype M2a, indicating a potential response to Pb-induced abnormalities in the forebrain cortex and cerebellum, aimed at brain repair. Up-regulation of the Chi3l1 and Mrc1 genes was observed in the forebrain cortex, while only up-regulation of the Chi3l1 gene was observed in the cerebellum. Thus, in the forebrain cortex, the anti-inflammatory and compensatory response of microglial cells seems to be the strongest. It is plausible that to reestablish homeostasis in the brain, microglial cells in the forebrain cortex and cerebellum to varying degrees may have attempted to switch from a pro-inflammatory M1 phenotype to a neuroprotective M2 phenotype during the progression of Pb-induced pathology. Observed in the forebrain cortex and cerebellum the attempt to change microglia phenotype to one that will allow for the repair and reconstruction of damaged tissue can be interpreted in two ways. This response may have been due to the wider range/spectrum of pathological changes or rather the specificity of the Pb-induced changes observed in the forebrain cortex and cerebellum compared to those in the hippocampus (e.g., unlike the hippocampus, only in the forebrain cortex and cerebellum we observed pathological changes in Tau protein; its excessive accumulation together with its hyperphosphorylation accompanied with GSK-3β and CDK5 kinases activation). We do not know to what extent Pb-induced hippocampal and cerebellar pathology in the Tau protein may determine phenotypic changes in microglia in these brain structures. In turn, taking into account both the degree of intensity of some pathological changes (oxidative stress and impairment of pro- and antioxidative balance of neurons; decrease in the number of neurons concomitantly with ultrastructural alterations) which were the most strongly observed in the hippocampus; and the wider spectrum of some abnormalities associated with synaptic pathology in the hippocampus and cerebellum (dyshomeostasis in a wider range of synaptic proteins, in both synaptic vesicles proteins (Syp, Syt1), presynaptic plasma membrane proteins (syntaxin1, SNAP25), and postsynaptic density proteins (PSD95) in contrast to changes only in PSD95 protein in forebrain cortex), we can speculate that the attempt to stimulate/switch from M1 to M2 phenotype in the forebrain cortex and also to a lesser extent in cerebellum however, it may be associated with a smaller range or their lesser intensity of pathological changes evoked by Pb in these brain regions. Perhaps both the nature and the degree of severity of the changes caused by Pb in the forebrain cortex and in the cerebellum, made it possible to attempt to activation of the mechanisms of neuroprotection (increase M2 genes expression). Alternatively, all these results may suggest the greater sensitivity of the hippocampus and cerebellum on Pb-induced pathology than the cortex. Nevertheless, all these data indicated that perinatal exposure to Pb induced long-lasting changes in the microglia status in a brain structure-specific manner. In the forebrain cortex and cerebellum, both a pro-inflammatory and a potentially beneficial recovery-promoting microglia phenotype were activated to a varying extent, while in the hippocampus, only an M1 phenotype was stimulated. To restore homeostasis in the brain, the microglia phenotype in the forebrain cortex and in the cerebellum likely made an attempt to switch from pro-inflammatory M1 to neuroprotective M2 during the pathology progression. Unfortunately, at this point we are unable to determine why the microglial response to Pb is specific to the brain regions studied. It is also unknown to what extent the activation of the two or one genes of the M2a phenotype of microglia is able to counteract Pb-induced pathology. However, despite the attempt to activate the M2 phenotype of microglia, we observed a wide range of neurotoxic effects of Pb on the brain (Figure 10), including pathological changes in the forebrain cortex and cerebellum. This evidence suggests rather an inability of microglia to counteract Pb-induced pathology and achieve a state of equilibrium in the brain. Although, in the case of the cortex, the range and intensity of pathology seem to be limited”.

We also prepared for the Reviewer Table S1 in which we showed the previously obtained results regarding the response of individual brain structures to Pb obtained on the same research model.

Table 1S. Brain regional vulnerability to Pb in the in model of perinatal exposure.

Abbreviation:  DYm - inner mitochondrial membrane potential; (AEC) Adenylate energy charge; (LPO) Lipid peroxides; (GSH) Reduced glutathione; (SOD1) copper/zinc superoxide dismutase; (SOD2) manganese superoxide dismutase; (CAT) catalase; (GPx1) glutathione peroxidase 1; (GPx4) glutathione peroxidase 4; (GSR) glutathione reductase; glycogen synthase kinase 3 Beta (GSK-3β); cyclin-dependent kinase 5 (CDK5); synaptophysin (Syp); synaptotagmin 1 (Syt-1); synaptosomal-associated protein, 25kDa (SNAP25); postsynaptic density protein-95 (PSD95)

Research model

Brain structure

Evaluated parameter

References

Primary cultures of granular neurons isolated from the cerebellum of rats subjected to perinatal exposure to Pb

Cerebellum

Impaired energy metabolism of neurons ↓

Mitochondria functions ↓

(DYm) ↓

Na+/K+ ATPase activity ↓

Adenylate energy charge (AEC) ↓

ATP, ADP, AMP concentration ↓

Baranowska-Bosiacka I. et.al. Altered energy status of primary cerebellar granule neuronal cultures from rats exposed to lead in the pre- and neonatal period. Toxicology, 2011, 280: 24–32.

In vivo model of Pb exposure in perinatal period.

Forebrain cortex (FC)

Cerebellum (C)

Hippocampus (H)

Oxidative stress and impairment of pro- and antioxidative balance of neurons in all examined brain structures:

Antioxidative enzymes cofactors concentration (Se, Cu, Mg, Zn) ↓

LPO concentration ↑

GSH concentration ↓ in FC and H

SOD1 activity ↓ in H

SOD2 activity ↓ in FC, C, H

CAT activity ↓ in C and H

GPx1 activity ↓ in FC

GPx4 activity unchanged in FC, C, H

GSR activity ↓ in FC and H

All examined brain structures showed impaired pro- and antioxidative balance, observed changes most strongly expressed in the hippocampus.

Baranowska-Bosiacka I. et al. Disrupted pro- and antioxidative balance as a mechanism of neurotoxicity induced by perinatal exposure to lead. Brain Res. 2012, 1435: 56-71.

In vivo model of Pb exposure in perinatal period.

Forebrain cortex (FC)

Cerebellum (C)

Hippocampus (H)

AIF expression ↓ in FC, C, H

Caspase activity unhanged in FC, C, H

Bax expression ↓ in FC, C, H

Bcl2 expression ↓ in FC, C, H

Bdnf expression ↓ in FC, C, H

All examined brain structures showed the reduction of the Bax/Bcl-2 ratio which reflect inhibition of apoptosis and could have been responsible for the lack of activation of Casp-3 and the lack of AIF translocation to the nucleus.

Decrease number of hippocampal neurons concomitantly with ultrastructural alterations in the hippocampus.

Baranowska-Bosiacka I. et al. Perinatal exposure to lead induces morphological, ultrastructural and molecular alterations in the hippocampus. Toxicology 2013, 303:187-200. 

In vivo model of Pb exposure in perinatal period.

Forebrain cortex (FC)

Cerebellum (C)

Hippocampus (H)

Tau pathology observed only in FC and C.

Increase in the phosphorylation of Tau at (Ser396) and (Ser199/202) with parallel rise in the level of total Tau protein in FC and C.

Tau hyperphosphorylation accompanied by elevated activity of GSK-3β and CDK5 kinases.

Activation of GSK-3β in FC and C activation of CDK5 in C.

We suggest that neurotoxic effect of Pb might be mediated, at least in part, by GSK-3β and CDK5-dependent Tau hyperphosphorylation, which may lead to the impairment of cytoskeleton stability and neuronal dysfunction.

GÄ…ssowska M. et al.

Perinatal exposure to lead (Pb) promotes Tau phosphorylation in the rat brain in a GSK-3β and CDK5 dependent manner: Relevance to neurological disorders. Toxicology,

Volumes 347–349,

2016, Pages 17-28.

In vivo model of Pb exposure in perinatal period.

Forebrain cortex (FC)

Cerebellum (C)

Hippocampus (H)

Pathological alterations within synapses observed  in all examined structures:

nerve endings swelling ↑,

enhanced packing density of

synaptic vesicles (SVs) in presynaptic area ↑ ,

blurred and thickened structure of the synaptic cleft.

In the vast majority of synapses the synaptic cleft was not visible and the postsynaptic density was blurred i.e. appears to be thickened and possess hardly discernible membranes.

Furthermore, in FC and H only few synaptic vesicles (SVs) were in contact

with presynaptic membrane.

The structural abnormalities were

accompanied by dyshomeostasis in the level of key synaptic proteins:

Syp ↑ in C

Syt-1 ↓ in C

SNAP25 ↓ in H

Syntaxin-1 ↓ in C, H

PSD-95 ↓ in FC, C

PSD95 ↑ in H

Ultrastructural altered mitochondria

(elongated, swollen or shrunken) observed in all examined structures.

Mitochondrial cristae and membrane

were fused and blurred.

GÄ…ssowska M.et al.  Perinatal exposure to lead (Pb) induces ultrastructural and molecular alterations in synapses of rat offspring. Toxicology. 2016 Dec 12; 373:13-29.

Point 6 In the discussion, the authors state that “Pb exposure exacerbates CNS damage by promoting M1 polarization and inhibiting M2 polarization of hippocampal microglia in rat brains” (lines 495-497). There is no data in this manuscript supporting an inhibition of M2 polarization state.

Answer: We would like to thank the Reviewer for this valuable comment. We fully agree with your opinion. We don’t have any data supporting an inhibition of M2 polarization state. This is mistake. Our data indicated the lack of changes in genes expression of M2 phenotype with concomitant strong cytotoxic phenotype M1 response in the hippocampus of Pb-exposed rats.  The lack of changes in gene expression of M2 phenotype may reflect the inability of this brain structure to restore homeostasis in response to wide scope pathological changes and abnormalities evoked by perinatal exposure to Pb. Taking into account the multi-directional toxicity action of Pb in the hippocampus, revealed by us in our previous studies, any attempt to change microglia phenotype to one that will allow for the repair and reconstruction of damaged tissue turns out to be ineffective. All these results may suggest also a greater sensitivity of the hippocampus on Pb-induced changes. Thus, we corrected the sentence: “Pb exposure exacerbates CNS damage by promoting M1 polarization and inhibiting M2 polarization of hippocampal microglia in rat brains” on " Pb exposure during fetal life and breastfeeding may exacerbate CNS damage by promoting M1 polarization of hippocampal microglia”.

Minor comments

-          Take care of the abbreviations, once defined you should use the abbreviated form. In particular, Pb or lead is used indistinctly.

We corrected the abbreviation in the whole manuscript according to Reviewer's remark.

Reviewer 2 Report

The authors have evaluated the neurotoxic effects of lead exposure on microglial and astrocyte activation in various brain specific regions in young rats. Lead exposure has been found to shift the population of microglial towards M1 phenotype resulting in an inflammatory cytokine storm which in turn is activates astrocytes. Lead exposure is also shown to cause synaptic vesicle dysregulation within the forebrain cortex. The study is highly relevant in terms of highlighting the detrimental effects of lead exposure in neonatal and early postnatal developmental stages. The paper needs to have some minor corrections.

1. Grammatical correction needed in some places

2. Introduction needs to be concise and to the point. The rationale of the study is submerged within the lengthy introduction.

3. Rearrange the figures according to each markers for better comprehension. For example, while discussing IL1B, group all figures relevant to this marker together from cortex, hippocampus and cerebellum in a single graph. 

4. Raw images for GS western not provided.  The Western images of GS and Gapdh looks similar to each other in Fig 6 A.D.  Please confirm the images are not repeated in Fig 6.

5. In the context of synaptic vessel at the presynaptic ends, please comment on any changes in the LTP or functions of the synaptic transmission.

Author Response

Dear Editor,

Dear Reviewers,

First of all, we would like to thank you very much for consideration of our manuscript: “Microglia and Astroglia – The Potential Role in Neuroinflammation Induced by Pre- and Neonatal Exposure to Lead (Pb)” for publications in the IJMS (Manuscript ID: IJMS-2383916). We have taken into consideration very carefully all remarks and comments of all  Reviewers. The manuscript was corrected according to the Reviewer’s remarks (all changes are marked in red colour in review).

The major changes suggested by Reviewers:

  • According to Reviewer suggestion the Introduction was shortened and more focused on the main issue of work. The main goal and conclusions were highlighted. All made changes are highlighted in red.
  • According to Reviewer suggestion the manuscript was enriched with additional immunohistochemical experiments showing the expression of Iba1 and GFAP in all analysed brain structures (Figure 1D and Figure 6D in revised manuscript).
  • According to Reviewer suggestion Discussion was corrected.
  • The manuscript was linguistically revised by a native English-speaking colleague.

We consider that all corrections increased the value of our article and our explanations you will find adequate. I hope our corrected manuscript can be accepted for publication in IJMS.

I would very much appreciate your opinion thereon.

Yours faithfully,

Irena Baranowska-Bosiacka

Answer to Reviewer 2

Reviewer #2: The authors have evaluated the neurotoxic effects of lead exposure on microglial and astrocyte activation in various brain specific regions in young rats. Lead exposure has been found to shift the population of microglial towards M1 phenotype resulting in an inflammatory cytokine storm which in turn is activates astrocytes. Lead exposure is also shown to cause synaptic vesicle dysregulation within the forebrain cortex. The study is highly relevant in terms of highlighting the detrimental effects of lead exposure in neonatal and early postnatal developmental stages. The paper needs to have some minor corrections.

  1. Grammatical correction needed in some places.
  2. Introduction needs to be concise and to the point. The rationale of the study is submerged within the lengthy introduction.
  3. Rearrange the figures according to each markers for better comprehension. For example, while discussing IL1B, group all figures relevant to this marker together from cortex, hippocampus and cerebellum in a single graph.
  4. Raw images for GS western not provided. The Western images of GS and Gapdh looks similar to each other in Fig 6 A.D. Please confirm the images are not repeated in Fig 6.
  5. In the context of synaptic vessel at the presynaptic ends, please comment on any changes in the LTP or functions of the synaptic transmission.

Answer: Thank you very much for the examination of our Manuscript and for valuable comments, and also for time offered for the preparation of revision. We have carefully taken into consideration all comments.

Point 1 Grammatical correction needed in some places.

Answer: The manuscript was linguistically revised by a native English-speaking colleague.

Point 2 Introduction needs to be concise and to the point. The rationale of the study is submerged within the lengthy introduction.

Answer: We would like to thank the Reviewer for this valuable comment. We fully agree with your opinion, so we shortened and modified the Introduction according to your suggestion. All changes to the Manuscript are highlighted.

Point 3 Rearrange the figures according to each markers for better comprehension. For example, while discussing IL1B, group all figures relevant to this marker together from cortex, hippocampus and cerebellum in a single graph.

Answer: Thank you very much for this remark. In our opinion, the presentation of our results grouping them due to the analyzed parameter (microglia’s response, expression of chemokines, changes in astrocytes phenotype, its activation and function, ultrastructural changes in glia cells) is clear and understandable and well illustrates the response of microglia and astrocytes to the action of Pb. Thus, we decided not to change the form of presenting the results.

Point 4 Raw images for GS western not provided. The Western images of GS and Gapdh looks similar to each other in Fig 6 A.D. Please confirm the images are not repeated in Fig 6.

Answer: According to your request, we added all original, untouched and uncropped Western Blot images. We checked all Western images and we confirm the images are not repeated and adequately represent the original images provided. All original images are uploaded as an attachment.

The signal was detected using the G BOX (Syngene), and the obtained images were saved using the GeneSnap program ver.7.12.01 (Syngene). This software does not combine both files (merge) to have the marker next to the signal. So we couldn’t take an image of both the ladder and the samples merged together in one image to see the position of the bands according to the ladder MW.

Point 5 In the context of synaptic vessel at the presynaptic ends, please comment on any changes in the LTP or functions of the synaptic transmission.

Answer: We would like to thank the Reviewer for this valuable comment. Indeed, in our model of pre- and neonatal Pb exposure, we observed an accumulation of synaptic vesicles (SVs) at the presynaptic ends. We paid particular attention to this phenomenon in our previous papers [Baranowska-Bosiacka 2013; GÄ…ssowska 2016]. The enhanced density of SVs in the presynaptic area together with other observed by us pathological changes within synapses can lead to synaptic dysfunction, expressed by the impairment of the secretory mechanism and thereby to the abnormalities in neurotransmission as well as to the neuronal dysfunction. Noted in our previous study defects in specific synaptic proteins may produce abnormalities in cellular neurotransmissions at the excitatory (E) and inhibitory (I) synapses, disrupting the E/I balance and affecting neural plasticity, which could be one of the fundamental causative factors underlying Pb neurotoxicity. The effect that Pb has on the brain causing cognitive impairment is related to its effects on dopaminergic, cholinergic and glutamatergic neurotransmission systems. Pb has been shown to suppress the Ca2+-related release of neurotransmitters: acetylcholine, dopamine, and amino acids [Cory-Slechta 2004]; also other authors indicate impaired storage and release of neurotransmitters [Braga 2004; Gill 2003; Lasley 1999]. Although the mechanism of this action of Pb is still not fully understood, it is known that it affects presynaptic channels involved in the release of neurotransmitters [Audesirk 1993] including NMDA channels [Lasley 1999]. Also in our previous studies, we demonstrated an inhibitory effect of Pb on NMDA channels [Gavazzo 2008]. Ubiquitous forms of long-term potentiation (LTP) and depression (LTD) are caused by enduring increases or decreases in neurotransmitter release. The effect that Pb has on the brain causing cognitive impairment is also associated with LTP impairment, which has so far been demonstrated in both in vitro and ex vivo acute poisoning conditions using tissues from chronically exposed animals [Cai 2001]. In in vivo models, Pb induced a decrease in the LTP amplitude and an increase in the LTP threshold [Zhu 2005]. The mechanism of LTP blockade was not clear in previous studies because the concentration of Pb that completely blocked LTP had no effect on the NMDA current. However, subsequent studies showed a significantly lower IC50 value at which Pb blocks the flow of the NMDA channel. [Hori 1993; Gavazzo 2001]. We discuss in detail the impact of Pb on the mechanism of LTP inhibition and the NMDA receptor in our review [Baranowska-Bosiacka 2006]. In the current manuscript, we wanted to focus mainly on pro-inflammatory changes in astrocytes and glial cells, therefore we did not study the phenomena associated with LTP. Unfortunately, at the current stage of research, we do not have such a possibility or equipment for electrophysiological tests.

References:

  • Baranowska-Bosiacka, I. et al. Perinatal exposure to lead induces morphological, ultrastructural and molecular alterations in the hippocampus. Toxicology 2013, 303, 187-200.
  • GÄ…ssowska, M. et al. Perinatal exposures to lead (Pb) induces ultrastructural and molecular alterations in the synapses of rat offspring. Toxicology 2016, 373, 13-29.
  • Cory-Slechta, D.A. et al. Maternal stress modulates the effects of developmental lead exposure. Environ Health Perspect 2004, 112:717-30.
  • Braga, M.F. et al. Pb2+ via protein kinase C inhibits nicotinic cholinergic modulation of synaptic transmission in the hippocampus. J Pharmacol Exp Ther 2004, 311:700-10.
  • Gill, K.D., Gupta, V., Sandhir, R. Ca2+/calmodulin-mediated neurotransmitter release and neurobehavioral deficits following lead exposure. Cell Biochem Funct 2003, 21:345-53.
  • Lasley, S.M. et al. Influence of exposure period on in vivo hippocampal glutamate and GABA release in rats chronically exposed to lead. Neurotoxicology 1999, 20:619-29.
  • Audesirk, G. Electrophysiology of lead intoxication: effects on voltage-sensitive ion channels. Neurotoxicology 1993, 14:137- 147.
  • Lasley, S.M. et al. Lead inhibits the rat N-methyl-d-aspartate receptor channel by binding to a site distinct from the zinc allosteric site. Toxicol Appl Pharmacol 1999, 59:224-33.
  • Gavazzo, P., Zanardi, I., Baranowska-Bosiacka, I., Marchetti, C. Molecular determinants of Pb2+ interaction with NMDA receptor channels. Neurochem Int. 2008, 52(1-2): 329-37.
  • Cai, L. et al. Effects of lead exposure on long-term potentiation induced by 2-deoxy-D-glucose in area CA1 of rat hippocampus in vitro. Neurotoxicol Teratol 2001, 23: 481-487.
  • Zhu, Z.W. et al. Study on the neurotoxic effects of low-level lead exposure in rats. J Zhejiang Univ Sci 2005, 6B:686-692.
  • Hori, N. et al. Lead blocks LTP by an action not at NMDA receptors. Exp Neurol 1993, 119:192-7.
  • Gavazzo, P. et al. Lead inhibition of NMDA channels in native and recombinant receptors. Neuroreport 2001, 12:3121-5.
  • Baranowska-Bosiacka, I. et al. Biochemical mechanisms of neurotoxic lead activity. Postepy Biochem. 2006, 52(3):320-9.

Round 2

Reviewer 1 Report

The authors have done a reasonable job covering most of the concerns raised by the reviewer and I appreciate it. Although introduction has been improved, I think it is still too long, need to be more concise.    

Author Response

Review 1

Comments and Suggestions for Authors

The authors have done a reasonable job covering most of the concerns raised by the reviewer and I appreciate it. Although introduction has been improved, I think it is still too long, need to be more concise.    

According to Reviewer remark we shortened Introduction section.